# ECO: Enhanced Code Optimization via Performance-Aware Prompting for Code-LLMs

## Abstract

Code runtime optimization—the task of rewriting a given code to a faster one—remains challenging, as it requires reasoning about performance trade-offs involving algorithmic and structural choices. Recent approaches employ code-LLMs with slow-fast code pairs provided as optimization guidance, but such pair-based methods obscure the causal factors of performance gains and often lead to superficial pattern imitation rather than genuine performance reasoning. We introduce ECO, a performance-aware prompting framework for code optimization. ECO first distills runtime optimization instructions (ROIs) from reference slow-fast code pairs; Each ROI describes root causes of inefficiency and the rationales that drive performance improvements. For a given input code, ECO in parallel employs (i) a symbolic advisor to produce a bottleneck diagnosis tailored to the code, and (ii) an ROI retriever to return related ROIs. These two outputs are then composed into a performance-aware prompt, providing actionable guidance for code-LLMs. ECO's prompts are model-agnostic, require no fine-tuning, and can be easily prepended to any code-LLM prompt. Our empirical studies highlight that ECO prompting significantly improves code-LLMs' ability to generate efficient code, achieving speedups of up to $7.81\times$ while minimizing correctness loss.

## 1 Introduction

Code runtime optimization—the task of rewriting a given code to a faster one—is a fundamental problem in software engineering, as it directly affects user experience and system performance (ISO/IEC, 2011). Recent advances in large language models for code (code-LLMs) demonstrated remarkable ability in ensuring functional correctness through tasks such as code synthesis, translation, and summarization (Chen et al., 2021; Xu et al., 2022). However, correctness alone does not imply efficiency; generating faster code requires performance-oriented reasoning that goes beyond code semantics. This gap makes code optimization particularly challenging for approaches that rely solely on the intrinsic capabilities of code-LLMs (Shypula et al., 2024).

Early works in code optimization utilized compiler-driven techniques, which applied rule-based analysis at the intermediate representation level, such as dead code elimination or loop unrolling (Wegman & Zadeck, 1991; Booshehri et al., 2013). These approaches are effective for addressing well-defined low-level inefficiencies, but they fail to capture the dominant performance bottlenecks—program-level, context-dependent optimizations including algorithmic restructuring or data-structure selection. Recent studies adopt code-LLMs to address this issue, with methods such as chain-of-thought (Wei et al., 2022) attempting to leverage their intrinsic reasoning ability. However, code-LLMs alone lack the capacity to optimize code and therefore require external guidance. Building on this, Shypula et al. (2024) and Gao et al. (2025) exploit slow-fast code pairs through prompting techniques such as in-context learning (ICL) and retrieval-augmented generation (RAG), where the example pairs are chosen randomly or by code-similarity retrieval. In parallel, fine-tuning approaches directly train models directly on slow-fast mapping.

Despite methodological diversity, existing LLM-based approaches shared a fundamental limitation: they guide models by presenting slow-fast code pairs as labeled transformation instances, which encourage pattern imitation rather than intent-aligned reasoning. Without interpretable guidance, the

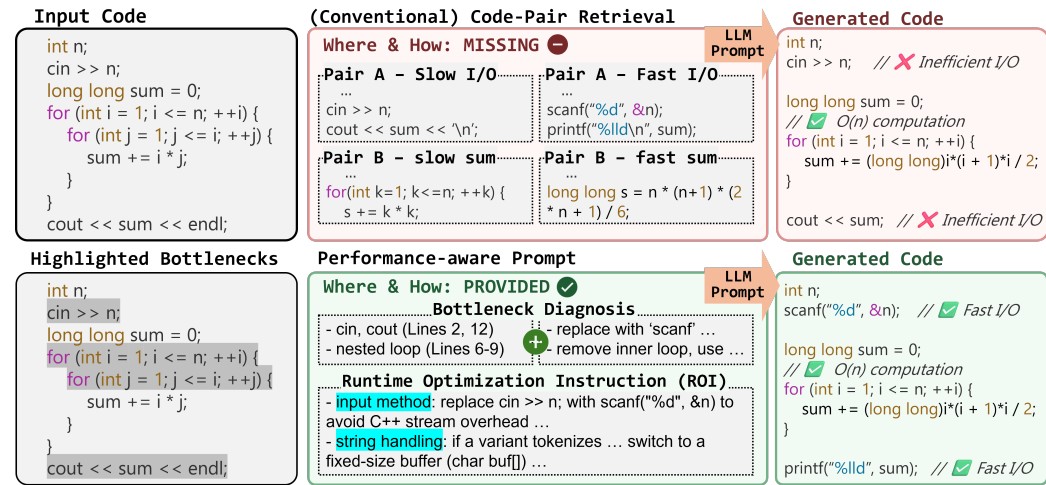

Figure 1: Comparison of ECO's performance-aware prompt with conventional retrieval.

model is left to infer why certain edits improve efficiency—a task that exceeds its intrinsic capabilities, so the guidance remains underutilized. As a result, retrieval often returns functionally similar snippets whose performance characteristics are misaligned with the target, and fine-tuned models tend to memorize recurring edits without recognizing their underlying rationale. This limitation suggests that code optimization requires more than exposing raw slow–fast pairs: models should instead be guided by performance-aware prompts that capture optimization intent.

In this context, we introduce ECO, a performance-aware prompting framework that provides code-LLMs with optimization insights tailored to the input code. Concretely, ECO first distills runtime optimization instructions (ROIs) from reference slow–fast code pairs—for example, identifying that a dynamic vector was replaced by a fixed-size array to reduce allocation overhead—and stores them as a knowledge base. At inference time, ECO leverages this knowledge to construct performance-aware prompts, combining complementary outputs from two modules: the symbolic advisor and the ROI retriever. Fig. 1 illustrates how ECO's performance-aware prompting leads to more effective optimizations.

The symbolic advisor runs graph-based queries over the input code's Code Property Graph (CPG) to deterministically identify structural inefficiencies and a bottleneck diagnosis—specifying where the bottlenecks lie and what type of transformation is required. In parallel, the ROI retriever retrieves performance-relevant ROIs grounded in prior examples, offering optimization instructions that generalize beyond fixed rules. Together, the two modules combine deterministic precision with contextual breadth, yielding performance-aware prompts that localize bottlenecks and prescribe transformations, which downstream code-LLMs can directly apply without fine-tuning.

We evaluate ECO on both the in-distribution PIE benchmark and the out-of-distribution Codeforces dataset, as well as across models of varying scales. ECO consistently achieves substantial runtime improvements while minimizing the loss in correctness. As model size and capacity grow, ECO fully utilizes the expanded capability, leading to state-of-the-art results. Due to its model-agnostic design, this property extends naturally to closed-source systems: for example, on *GPT-o4-mini*, standard prompting yields only a $1.99\times$ speedup, whereas ECO raises to $7.81\times$—nearly a four-fold gain. These results underscore the effectiveness and practicality of ECO's performance-aware prompting.

Our key contributions are:

- Performance-aware prompting: We move beyond raw slow-fast code pairs by distilling runtime optimization instructions and composing them into performance-aware prompts.
- Complementary module design: We design a rule-based symbolic advisor for deterministic bottleneck detection and an ROI retriever for context-aware, generalizable guides.
- Model-agnostic, plug-in framework: ECO requires no fine-tuning or model-specific adaptation. Its directives integrate seamlessly into the prompt of any code-LLM.

Table 1: Characteristics of code optimization methods. Upper rows correspond to generic LLM approaches, lower rows correspond to code optimization-oriented methods. *Optimization Knowledge* denotes the underlying origin that drives the guidance.

| | Program Scope | Train-Free | Non-Iterative | Bottleneck Diagnosis | Optimization Knowledge |
|---|---|---|---|---|---|
| Intruction-only | ○ | ○ | ○ | × | none |
| ICL | ○ | ○ | ○ | × | code-pair |
| RAG | ○ | ○ | ○ | × | code-pair |
| Fine-tune (PIE) | ○ | × | ○ | × | code-pair |
| Compiler | × | ○ | ○ | × | rule-based |
| Supersonic | ○ | × | ○ | × | diff patch |
| SBLLM | ○ | ○ | × | × | code-pair |
| ECO (Ours) | ○ | ○ | ○ | ○ | ROI |

## 2 RELATED WORKS

Traditional compiler infrastructures such as LLVM (Lattner & Adve, 2004) and GCC (Free Software Foundation, 2025) apply rule-based optimizations at the intermediate representation (IR) level. While effective at eliminating low-level inefficiencies, these techniques remain limited in addressing program-level, context-dependent bottlenecks, including algorithmic restructuring or data-structure selection. This limitation motivated the emergence of approaches that leverage code-LLMs.

PIE (Shypula et al., 2024) explored this direction by fine-tuning models on slow-fast code pairs, while Supersonic (Chen et al., 2024) extended this paradigm by training to predict the edit operations (i.e., diff patches) that lead to its optimized counterpart, rather than raw code pairs. Such approaches require costly retraining for model update and provide limited gains, as the supervision essentially encodes before-after patterns without exposing the underlying rationale for runtime improvements. In contrast, SBLLM (Gao et al., 2025) adopted a retrieval-augmented prompting (RAG) combined with iterative revision. At each step, the system retrieves the code pair most similar to the current candidate in embedding space and uses it to guide the code-LLM in refining candidates. This process increases inference cost and often leads to semantic drift, where repeated rewriting gradually alters or even breaks the original functionality of the code.

Existing LLM-based approaches typically draw on past optimization instances directly—such as code pairs or diff patches—which present surface-level transformations without translating them into performance-aware signals. As a result, they do not provide bottleneck diagnoses or any direct optimization guidance, leaving models to infer performance bottlenecks on their own and often producing suboptimal fixes. In contrast, ECO introduces runtime optimization instructions (ROIs) and delivers bottleneck diagnoses, supplying richer and more targeted information for optimization. In this way, ECO directly addresses the lack of precise diagnoses and actionable knowledge that limits prior methods.

## 3 PROPOSED FRAMEWORK

We introduce ECO, a performance-aware prompting framework. Given input codes to be optimized, ECO generates performance-aware prompts, offering actionable hints that the model can immediately act upon. Unlike prior approaches that merely expose slow–fast code pairs and leave code-LLMs to implicitly infer optimization patterns, ECO directly provides two complementary forms of guidance: (i) a bottleneck diagnosis that pinpoints where inefficiencies occur and what type of transformation is required, and (ii) related runtime optimization instructions (ROIs) from past optimizations that offer concrete, performance-relevant examples. This design allows code-LLMs to bypass the effort of discovering bottlenecks themselves and instead focus on applying the suggested transformations. Moreover, ECO operates in a model-agnostic manner: its prompts can be simply prepended to any code-LLM input, requiring no fine-tuning or model-specific adaptation.

Figure 2 provides an overview of ECO. It builds on an ROI distillation step, where ROIs are extracted from reference slow-fast code pairs and stored in an ROI DB. This repository serves as the foundation for two modules. The symbolic advisor produces bottleneck diagnoses by applying graph-

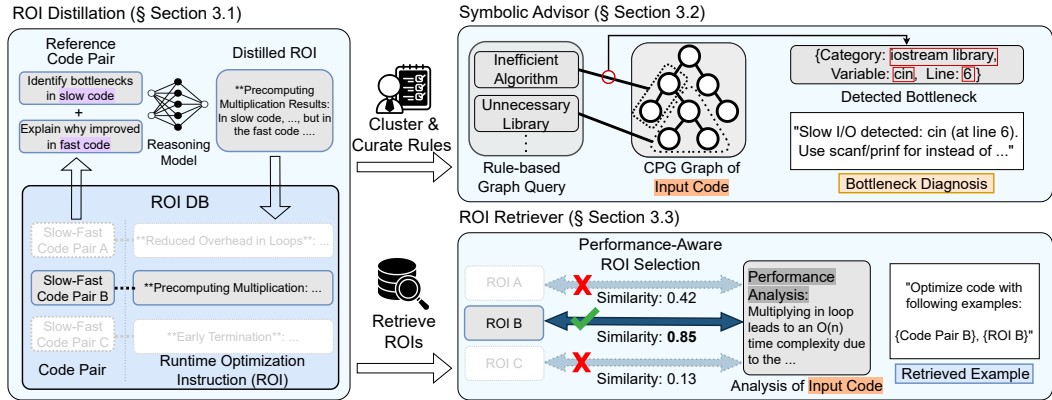

Figure 2: Overview of our ECO framework. ECO locates bottleneck code snippets using the symbolic advisor and the strategy retriever module, and then generates directives for code-LLMs.

based rules derived from clustered ROIs, deterministically identifying inefficiency patterns in the input code. In parallel, the ROI retriever matches an input-conditioned analysis against the database to return related ROIs, supplying broad-coverage optimization knowledge that extends beyond fixed rules. Together, these modules transform distilled ROI knowledge into performance-aware prompts that effectively steer downstream code-LLMs.

## 3.1 ROI DISTILLATION

The first step of ECO is to construct a database of runtime optimization instructions (ROIs) that can serve as prior knowledge for later performance-aware prompting. We use the PIE HQ dataset with 4,085 slow-fast code pairs, each solving the same problem but exhibiting substantial runtime differences. ECO leverages these examples to distill generalizable optimization knowledge. Rather than consuming raw pairs directly, ECO abstracts them into ROIs, which capture not only what changed but also why the changes improve efficiency.

For each code pair $i$, we prompt a reasoning-oriented LLM to analyze the slow and fast implementations and to extract a compact ROI $O_i$. The prompting design encourages the model to reason explicitly about the root cause of inefficiency and producing a compact natural-language instruction $O_i$. We then build an ROI database of triplets that link each slow–fast code pair with its distilled ROI,

$$\mathcal{D} = \{(\text{slow}_i, \text{fast}_i, O_i)\}.$$

This corpus serves as ECO's knowledge base, effectively transforming raw code pairs into structured optimization knowledge that can be systematically reused across models and tasks. Details of the prompting template and LLM configuration are in Appendix A.1.

## 3.2 SYMBOLIC ADVISOR

The symbolic advisor applies rule-based queries to capture deterministic inefficiency patterns and generates bottleneck diagnoses with matched templates. Built on top of *Joern*[1], which constructs Code Property Graphs (CPGs) from source code, we design carefully crafted graph queries and explanatory templates that detect inefficient program entities and articulate how to address them. These rules and templates are grounded in the ROI database distilled in Section 3.1. We manually cluster similar ROIs and translate each cluster into formal rule-template pairs, enabling the symbolic advisor to convert distilled knowledge into precise, program-level bottleneck diagnoses. Unlike linters (GmbH, Accessed: 2025-05-31) that only match shallow syntax patterns or compiler optimizations (Free Software Foundation, 2025; Lattner & Adve, 2004) that operate on low-level intermediate representations, our approach enables program-level structural analysis—such as identifying redundant recursion through call-graph inspection or detecting dynamic containers that do not exploit resizing by tracing object operations.

---

[1] https://joern.io/

The symbolic advisor operates on a set of rule–template pairs $\mathcal{P} = \{(r_1, t_1), \ldots, (r_m, t_m)\}$, where each rule $r$ is a Joern query that isolates an inefficiency pattern and the associated template $t$ specifies how to verbalize the match into a natural language directive. Concretely, the rule set covers four categories of inefficiency: (i) Inefficient Algorithms, (ii) Data Structure Usage, (iii) Library Usage, and (iv) Loop Structures. Detailed examples of these categories are provided in Appendix A.2.

---

**Algorithm 1** Detect Recursion Methods without Memoization

1: **Input:** Code Property Graph $G$
2: **Output:** Matches $\mathcal{M}$     ▷ set of recursive methods w/o memoization

3: $\mathcal{S} \leftarrow$ SELFCALLMETHODS$(G)$
4: **for** each $f$ in S **do**
5:     $\mathcal{R}_f \leftarrow$ INDIRECTREADS$(f)$
6:     $\mathcal{W}_f \leftarrow$ INDIRECTWRITES$(f)$
7:     $\mathcal{C}_f \leftarrow \mathcal{R}_f \cap \mathcal{W}_f$
8: **end for**
9: $\mathcal{M} \leftarrow \{f \in$ S $\mid \exists id \in \mathcal{C}_f \wedge \neg$DECLARES$(f, id)\}$
10: **return** $\mathcal{M}$

---

**Algorithm 2** Symbolic Advisor Pipeline

1: **Input:** Input code $C$, rule–template pairs $\mathcal{P} = \{(r_1, t_1), \ldots, (r_m, t_m)\}$
2: **Output:** Bottleneck Diagnoses $\mathcal{B}$

3: $G \leftarrow$ BUILDCPG$(C)$     ▷ returns a CPG
4: $\mathcal{B} \leftarrow \emptyset$
5: **for** each $(r, t) \in \mathcal{P}$ **do**
6:     $\mathcal{M}_r \leftarrow r(G)$
7:     **for** each match $m \in \mathcal{M}_r$ **do**
8:        $b \leftarrow$ INSTANTIATE$(t, m)$
9:        $\mathcal{B} \leftarrow \mathcal{B} \cup \{b\}$
10:     **end for**
11: **end for**
12: **return** $\mathcal{B}$

---

As an example, Algorithm 1 defines a rule in 'Inefficient Algorithm' category to detect recursive functions that recompute overlapping subproblems without memoization. The rule first identifies all self-call methods from the given CPG (line 3), then collects identifiers that are both read and written inside each function (line 4-8), and finally checks whether these identifiers are declared as memoization tables (line 9). Functions without such declarations are flagged as inefficient recursive implementations. These rules are implemented in Scala using Joern's query language, and the complete code are provided in our public repository.

Algorithm 2 summarizes the overall pipeline of the symbolic advisor. Given an input code $C$, it builds a CPG $G$ (line 3), applies each rule in $\mathcal{P}$ to extract matches (line 5-6), and instantiates the corresponding templates into bottlenecks diagnoses (line 7-10). The resulting set $\mathcal{B}$ specifies inefficiency patterns and prescribes corresponding directives for improvement. In practice, the symbolic advisor reliably identifies deterministic inefficiencies but delegates concrete code transformations to the LLM refinement stage.

### 3.3 ROI RETRIEVER

The ROI retriever complements the symbolic advisor by supplying performance-aware guidance from past optimizations. Whereas traditional retrieval-augmented methods rely on code embeddings capturing only syntactic or semantic similarity, our retriever instead extracts a performance-aware description of the input code and matches it against distilled runtime optimization instructions (ROIs). First, this makes retrieval explicitly performance-relevant: the returned cases are aligned with the input code's performance characteristics rather than merely resembling it at the surface level. Second, beyond providing the slow-fast code pair itself, the ROI retriever also delivers the corresponding ROI, giving the model an explicit description of the inefficiency and its remedy.

Formally, ROI retriever builds on the ROI database $\mathcal{D}$ distilled in Section 3.1, where each entry $(\text{slow}_i, \text{fast}_i, O_i)$ is paired with a vector representation $\mathbf{v}_i = \Phi(O_i)$ produced by an embedding model $\Phi$. At inference time, given an input code $C$, we prompt the inference model itself with a structured query asking it to describe $C$'s performance-related characteristics. For a practical on-line setting, we reuse the inference model instead of invoking a separate reasoning model. This prompt is designed to mirror the ROI distillation step, so that the input is analyzed under a similar reasoning environment. The resulting explanation $E_C$ is then embedded as $\mathbf{v}_C = \Phi(E_C)$ and compared against all $\mathbf{v}_i \in \mathcal{D}$. The retriever selects the top-$k$ most similar entries and returns their associated triples $(\text{slow}_i, \text{fast}_i, O_i)$. This provides the LLM with not only relevant code examples but also richer and more interpretable ROI descriptions, offering accessible and performance-aware prompts.

# 4 EMPIRICAL STUDIES

## 4.1 EXPERIMENTAL SETTINGS

### 4.1.1 MODELS

ECO employs three types of models with distinct roles. (i) the reasoning model distills optimization strategies from slow–fast code pairs to construct the ROI database; we adopt *DeepSeek-r1:32b*. (ii) the embedding model vectorizes both code and instructions for similarity-based retrieval in the instruction retrieve; we adopt *Qodo-Embed-1-1:5b*, designed to capture both code and natural language characteristics. (iii) the inference model serves as the downstream code-LLM that refines the input code under the provided directives. We adopt the *Qwen2.5-Coder* family for open-source models, and *GPT-4o-mini* and *GPT-o4-mini* for closed-source models with the temperature 0.7.

### 4.1.2 BASELINE METHODS

The following methods were originally designed as generic LLM prompting techniques and later adapted to code optimization in PIE (Shypula et al., 2024). The instruction-only method provides standard instruction prompts to perform optimization without any additional guidance. Chain-of-Thought (CoT) (Wei et al., 2022) augments prompts by encouraging explicit step-by-step reasoning. In-Context Learning (ICL) (Brown et al., 2020) supplies randomly selected slow-fast code pairs, while dynamic retrieval (RAG) (Poesia et al., 2021) supplies pairs based on code-to-code similarity. However, RAG primarily captures syntactic resemblance rather than reflecting optimization intent.

The following methods are explicitly designed for code optimization. Supersonic (Chen et al., 2024) employs CodeBERT (Feng et al., 2020), an encoder model rather than an LLM, and trains it to generate refinement patches that transform a given slow code into its optimized fast version. SBLLM (Gao et al., 2025) combines a search-based approach with RAG. Starting from an initial program, it iteratively scores candidates, retrieves examples via code-level similarity, and applies a code-LLM to generate its optimized version. Detailed implementations of methods are provided in Appendix B.

### 4.1.3 DATASET

We use the widely adopted Performance Improvement Edits (PIE) C++ benchmark, which originates from CodeNet Puri et al. (2021). For training and reference, we leverage its high-quality set (HQ) consisting of 4,085 slow-fast code pairs, which is used for tasks such as fine-tuning, strategy distillation, and retrieval in both baseline methods and ECO. For evaluation, the PIE test set suffers from severe imbalance in problem distribution; we construct a balanced subset of 255 slow codes, each accompanied by 10 test cases. For assessing out-of-distribution (OOD) generalization, we additionally curate the Codeforce C++ dataset, from which we sample 300 slow codes, each also paired with 10 test cases. Details on dataset construction are provided in Appendix C.

### 4.1.4 EVALUATION METRICS

We use three standard evaluation metrics to assess both optimization effectiveness and functional correctness. We follow the best@$k$ protocol—selecting one candidate with the highest speedup from the $k$ generated candidates. Given original and optimized runtime $T(o)$ and $T(n)$ of a code,

- Percent optimized (OPT): The percentage of solutions that are both correct and at least 10% faster than the original code, i.e., $T(o) - T(n) > 0.1 \times T(o)$.

- Speedup rate (SP): The runtime reduction ratio, defined as $\text{SP} = T(o)/T(n)$. If the optimized code is incorrect or slower, we assign $\text{SP} = 1.0$.

- Accuracy (ACC): The percentage of optimized codes that are functionally equivalent to the original code, i.e., pass all provided test cases.

We compile all C++ programs using GCC 9.3.0 with the C++17 and the $-\text{O3}$ flag. Reported performance gains exclude compiler-level optimizations and reflect improvements beyond the compile-time baseline. We measure code runtime by employing Gem5 Binkert et al. (2011), a cycle-accurate system simulator that provides deterministic measurements essential for reliable benchmarking.

Table 2: Average performance of baseline methods and ECO, reported with standard deviations over 10 trials. We use *Qwen2.5-coder:7b* as an inference model.

| Methods | Best@1 | | | Best@5 | | |
|---|---|---|---|---|---|---|
| | ACC(%) | SP | OPT(%) | ACC(%) | SP | OPT(%) |
| Instruction-only | 33.61 (±2.18) | 1.17× (±0.06) | 5.92 (±1.41) | 68.12 (±1.54) | 1.44× (±0.03) | 15.92 (±1.20) |
| CoT (Wei et al., 2022) | 34.67 (±1.44) | 1.16× (±0.03) | 6.04 (±0.44) | 63.61 (±1.21) | 1.39× (±0.02) | 15.18 (±0.79) |
| ICL (Brown et al., 2020) | 35.33 (±2.00) | 1.27× (±0.05) | 8.12 (±1.22) | 70.75 (±1.24) | 1.82× (±0.04) | 23.10 (±0.94) |
| RAG (Poesia et al., 2021) | 29.69 (±3.01) | 1.52× (±0.15) | 11.41 (±1.43) | 64.51 (±1.78) | 2.51× (±0.12) | 30.31 (±1.12) |
| Supersonic (Chen et al., 2024) | 7.06 (±3.63) | 1.00× (±0.01) | 0.04 (±0.12) | 14.75 (±0.61) | 1.01× (±0.01) | 0.20 (±0.20) |
| SBLLM (Gao et al., 2025) | 21.73 (±2.21) | 1.06× (±0.01) | 2.35 (±0.53) | 55.80 (±3.15) | 1.22× (±0.04) | 7.61 (±0.95) |
| ECO | 36.27 (±2.88) | 2.15× (±0.11) | 23.84 (±1.13) | 74.24 (±1.46) | 3.26× (±0.09) | 48.04 (±1.17) |

## 4.2 COMPARISON WITH BASELINES

We evaluate ECO and baselines on the PIE dataset to assess their ability in code optimization. As shown in Table 2, ECO attains the highest SP and OPT with minimal loss in correctness. This demonstrates that ECO, by providing direct bottleneck diagnoses and relevant ROIs, is more effective than approaches that rely solely on raw slow–fast code pairs. Notably, ECO attains these gains without any additional model training or iterative search, underscoring the importance of supplying performance-aware information through carefully designed prompts.

Instruction-only and CoT achieve low performance as they provide no external guidance, relying solely on the intrinsic ability of code-LLMs. In contrast, ICL and RAG supply raw slow–fast code pairs directly as guidance—either randomly or based on code-level similarity. While this helps more than unguided prompting, such examples still fail to align with the actual performance bottlenecks of the input code. Our ECO, instead, retrieves examples by matching performance characteristics between the given code and candidates, and further augments them with distilled ROIs, thereby providing aligned and informative guidance.

Interestingly, Supersonic and SBLLM show even lower speedup than generic baselines, primarily due to their very low accuracy. Supersonic trains model to output the code differences (i.e., diff patches) between slow and fast code. However, this formulation frequently results in invalid outputs, with over 80% of produced patches as malformed. SBLLM employs an iterative search process that modifies the code over multiple steps, but this iterative nature increases the risk of semantic degradation. We interpret these results as evidence that both methods fail to leverage recent code-LLMs effectively and instead suffer from additional accuracy overhead.

Table 3: Average performance of ECO variants, reported with standard deviations over 10 trials. We use *Qwen2.5-coder:7b* as an inference model. The variants are denoted as ECO with ablations: SA refers to the symbolic advisor and RR to the ROI retriever. The best results are in ▓▓▓▓ and the second-best are in ▓▓▓ .

| Methods | Best@1 | | | Best@5 | | |
|---|---|---|---|---|---|---|
| | ACC(%) | SP | OPT(%) | ACC(%) | SP | OPT(%) |
| ECO | 36.27 (±2.88) | 2.15× (±0.11) | 23.84 (±1.13) | 74.24 (±1.46) | 3.26× (±0.09) | 48.04 (±1.17) |
| w/o RR | 48.59 (±1.47) | 1.97× (±0.14) | 21.61 (±1.10) | 83.45 (±1.29) | 3.08× (±0.10) | 42.75 (±1.19) |
| w/o SA | 32.98 (±2.14) | 1.87× (±0.15) | 18.39 (±2.06) | 70.39 (±2.00) | 3.10× (±0.12) | 42.12 (±1.11) |
| w/o RR+SA | 36.20 (±2.04) | 1.38× (±0.13) | 9.41 (±1.58) | 73.80 (±1.79) | 2.26× (±0.16) | 30.75 (±2.49) |

## 4.3 ABLATION STUDIES: ROLE OF SUBMODULES IN ECO

We conduct ablation study in Tab. 3. ECO without both modules essentially behaves like RAG: it selects reference code pairs by code similarity. The only addition is that it also provides the corresponding ROIs extracted from those pairs. Interestingly, its performance is slightly lower than RAG, even though it provides more information. This suggests that ROIs associated with code-level similar pairs may not align with the actual performance bottlenecks of the given input and can even introduce mismatches that hinder optimization. This highlights that merely supplying ungrounded ROIs is insufficient; the key is to select relevant ROIs and transform them into explicit prompts.

Both ECO variants with a single module individually surpass all baseline methods, showing that either module alone already provides helpful guidance than prior approaches. The symbolic advisor variant achieves the highest correctness due to its deterministic, rule-based nature. In contrast, the ROI retriever variant explores diverse optimization scenarios through retrieval, yielding comparable speedup but relatively lower correctness due to its exploratory search. When combined, ECO softens the trade-off in correctness (ACC 74.24%) while further enhancing speedup (SP 3.26×) in Best@5, indicating that each module compensates for the other's blind spots and yields the best optimization.

Table 4: Performance of ECO evaluated on the in-distribution PIE benchmark and the out-of-distribution Codeforce benchmark, using models of different scales and including closed-source models. The results demonstrate that ECO generalizes effectively across both models and datasets.

| Model | Prompting | PIE (Best@5) | | | Codeforce (Best@5) | | |
|---|---|---|---|---|---|---|---|
| | | ACC(%) | SP | OPT(%) | ACC(%) | SP | OPT(%) |
| Qwen2.5-Coder:3b | Instruction-only | 55.22 | 1.30× | 11.18 | 18.87 | 1.01× | 0.40 |
| | ECO | 37.80 | 1.85× | 16.67 | 16.17 | 1.11× | 2.57 |
| Qwen2.5-Coder:7b | Instruction-only | 68.12 | 1.44× | 15.92 | 29.20 | 1.01× | 0.47 |
| | ECO | 74.24 | 3.26× | 48.04 | 35.20 | 1.78× | 13.83 |
| Qwen2.5-Coder:14b | Instruction-only | 67.76 | 1.48× | 17.92 | 39.50 | 1.09× | 2.13 |
| | ECO | 79.84 | 3.67× | 53.69 | 45.40 | 2.30× | 23.83 |
| GPT-4o-mini | Instruction-only | 83.53 | 1.53× | 19.61 | 49.23 | 1.01× | 0.33 |
| | ECO | 94.51 | 3.97× | 60.78 | 59.93 | 2.01× | 18.70 |
| GPT-o4-mini | Instruction-only | 95.29 | 1.99× | 36.08 | 65.63 | 1.41× | 7.33 |
| | ECO | 97.25 | 7.81× | 84.71 | 73.67 | 4.55× | 42.07 |

## 4.4 GENERALIZABILITY OF ECO

As shown in Table 4, we evaluate ECO on *Qwen2.5-coder* models of different sizes. On the smallest 3B model, applying ECO decreases accuracy: with limited capacity, the model often fails to faithfully implement the provided guidance, whereas instruction-only prompting makes fewer edits and thus retains higher correctness. As model capacity increases, however, our strategy-driven guidance takes full advantage of stronger reasoning ability—both accuracy and speedup rise substantially, with the 14B model achieving the largest gains.

We conduct experiments on closed-source LLMs, *GPT-4o-mini* (general-purpose) and *GPT-o4-mini* (reasoning-oriented). ECO can be used as plug-in guidance without any fine-tuning or model-specific adaptation, underscoring its model-agnostic design. Instruction-only prompting yields only marginal speedups under 2×, which is the standard way of utilizing LLMs, whereas ECO improves substantially, with *GPT-o4-mini* achieving a remarkable **7.81×** speedup on the PIE dataset. These results indicate that ECO is not tied to any specific model and can be readily applied to new LLMs as they emerge. They further suggest that when a higher-capacity model is supplied with explicit, high-quality guidance, it can faithfully follow it and achieve significant performance improvements.

We assess ECO's robustness under out-of-distribution (OOD) conditions using the more challenging Codeforces benchmark. While ECO's ROIs are distilled from the PIE HQ dataset, the OOD setting introduces problems whose performance characteristics differ from the training distribution. This benchmark is highly difficult: under instruction-only prompting, all models except *GPT-o4-mini* achieved speedups below 1.1×. In contrast, applying ECO guidance in the Codeforce setting leads to substantial improvements across models, with the most dramatic case showing a jump from 1.01× to 2.01× speedup. These results demonstrate that ECO provides robust guidance even under OOD conditions and enables significant gains on difficult optimization tasks.

## 4.5 CASE STUDY

Fig. 3 shows the distribution of ECO's outputs across single-trial generations, largely categorized into optimized, correct but not optimized, and failed. Compared to baselines, ECO yields a much higher proportion of optimized outputs, while the share of 'correct but not optimized' results is relatively lower. This trade-off arises from the stochastic nature of the ROI retriever. Although it

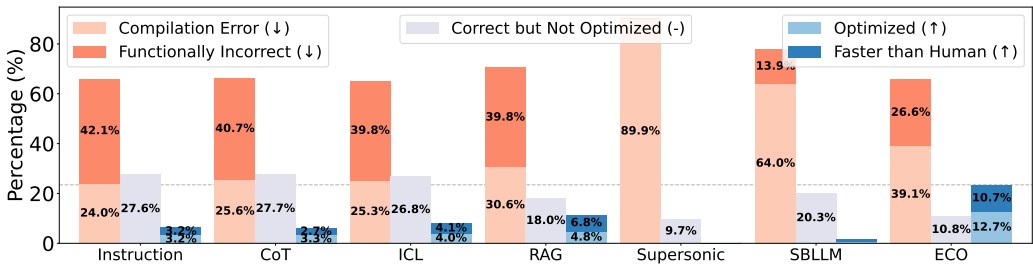

Figure 3: Detailed analysis of errors (↓) and correct but not optimized (-), and optimized (↑) cases for each method. Lower error percentages and higher optimized percentages are better.

sometimes selects less relevant examples, it also enables stronger optimizations when the match is effective, leading to variability. Allowing multiple generations (e.g., Best@5) mitigates this issue by increasing the chance of retrieving more relevant ROIs. In contrast, Instruction-only, ICL, and RAG produce a larger fraction of 'correct but not optimized' outputs, indicating that they often fail to identify meaningful optimization opportunities.

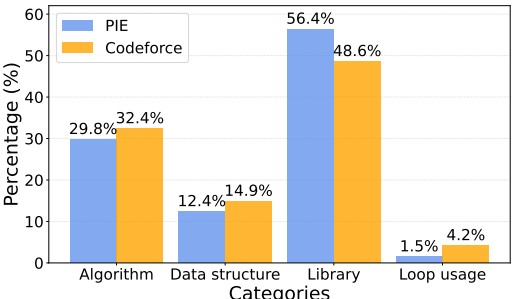

| Rank | RAG | | ROI retriever | |
| --- | --- | --- | --- | --- |
| | Keyword | TF-IDF | Keyword | TF-IDF |
| 1 | sum | 0.4127 | ans | 0.2432 |
| 2 | mid | 0.3761 | string | 0.2350 |
| 3 | cnt | 0.2743 | long | 0.1778 |
| 4 | rep | 0.2352 | max | 0.1270 |
| 5 | ans | 0.2298 | scanf | 0.1142 |
| 6 | long | 0.1727 | cin | 0.1119 |
| 7 | size | 0.1599 | cout | 0.1055 |
| 8 | string | 0.1511 | vector | 0.1028 |
| 9 | double | 0.1497 | endl | 0.1024 |
| 10 | mod | 0.1434 | sort | 0.0883 |

Figure 4: (Left) Distribution of bottlenecks detected by the symbolic advisor. (Right) TF-IDF ranking of the top-10 overlapping keywords between input and retrieved code, under RAG and ROI retriever on the PIE test set.

As shown in the left side of Fig. 4, we apply the symbolic advisor's detection rules to the test sets to analyze the distribution of performance bottlenecks. The majority of issues arise from library usage, with I/O inefficiencies being the most prominent. Algorithm-level inefficiencies also constitute a large portion of bottlenecks. By contrast, data structure misuse and inefficient loop usage appear less frequently. These findings show that the dominant bottlenecks lie in categories requiring semantic-level reasoning beyond low-level code patterns.

The right table in Fig. 4 presents a comparison of overlapping keywords between input code and retrieved code under two retrieval methods. For each input-retrieved code pair, we identify the overlapping keywords and then weight them by their average TF-IDF scores. Notably, in the code retrieved by our ROI retriever, the most frequently overlapping keywords are performance-relevant terms, such as I/O operations (e.g., scanf, cin) and data-structure tokens (e.g., vector, sort). In contrast, the RAG, code-similarity retrieval, is dominated by superficial overlaps, such as variable names (e.g., cnt, ans) or arithmetic-related tokens (e.g., sum, mod).

## 5 CONCLUSIONS

We propose ECO, a performance-aware prompting framework for code optimization. ECO distills runtime optimization instructions (ROIs) from reference code pairs and, for a given input, produces performance-aware prompts by combining a bottleneck diagnosis with related ROIs. These prompts are model-agnostic and require no fine-tuning, yet significantly improve runtime efficiency across diverse LLMs. Our results underscore that providing explicit performance-aware prompts is a practical and effective approach for enabling code-LLMs to generate optimized code.

REPRODUCIBILITY STATEMENT

We provide extensive details to ensure reproducibility. The prompts used for code-LLMs are given in Appendix A. Appendix B further describes the inference model configuration, baseline implementations, and measurement tools. The dataset curation process is explained in Appendix 4.1.3. Finally, we release the supplementary code repository, which includes the full implementation of ECO and the curated dataset for easy replication.

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

# A    DETAIL OF ECO

## A.1    ROI DISTILLATION

We utilize the PIE HQ dataset comprising 4,085 slow-fast code pairs, where each pair addresses the same problem yet exhibits a substantial runtime difference. From these reference code pairs, ECO extracts runtime optimization instructions (ROIs) using a reasoning-oriented LLM, *DeepSeek-r1:32b*, executed through the ollama framework. In our setup, the model runs in a quantized configuration (Q4_K_M), a 4-bit quantization scheme. Specifically, given a slow code and a fast code, we prompt the model with a manually crafted template to extract optimization objects. Each object explains the inefficiency in the slow implementation, how the fast implementation addresses it, the estimated runtime impact, and its category. The prompt template is shown below:

```
Prompt template for ROI distillation

Identify each optimization in the Slow Code
and explain how it speeds up the Fast Code.
Respond in JSON array form, with objects containing:
[
  {
    "description": "Briefly describe the inefficiency in slow_code and
    how fast_code fixes it.",
    "runtime_improvement": "Integer (1-10) rating of runtime gain.",
    "category": "One of: Algorithm | Data Structure | Memory Management
    | Code Execution | System Interaction | Other"
  },
  ...
]
Slow Code:
{slow_code}

Fast Code:
{fast_code}
```

From the model's raw output, we extract only the compact optimization instruction $O_i$, which typically appears after the </think> marker that separates reasoning steps. An example of such instruction is shown in Fig. 5. We then store each resulting triplet—consisting the slow code, fast code, and the extracted instruction—in a ROI database, formally represented as $\mathcal{D} = \{(\text{slow}_i, \text{fast}_i, O_i)\}$. These instructions subsequently serve as the knowledge base for ECO's modules, enabling performance-oriented prompting.

## A.2    SYMBOLIC ADVISOR

In the symbolic advisor, we leverage Joern, a tool that converts code into graph representations such as ASTs and code property graphs (CPGs). Joern enables static analysis through its rich graph query language, which supports function call exploration and various structural queries. Based on the optimization instructions extracted earlier, we manually clustered them and distilled knowledge into four major categories, each represented by a set of rules implemented in Scala. Each rule checks whether specific nodes satisfying certain conditions exist in the code graph. Depending on the rule, we then generate explicit templates that specify where and how to optimize, producing performance-aware prompts accordingly. The implementation is written in Scala and released in the supplementary material. This process is fully deterministic and independent of any code-LLM.

---

**Optimization instruction example**

The optimization points between the slow and fast code can be identified as follows:

1. **Replacing 'cout' with 'printf':**
   − The slow code uses 'cout', which involves more overhead due to its object−oriented nature, while the fast code uses 'printf', a function from the C standard library that is more efficient for I/O operations.

2. **Precomputing Multiplication Result:**
   − In the slow code, the multiplication is done inline within the output statement, whereas in the fast code, it's precomputed and stored in a variable ('mt'). This avoids recalculating the result multiple times.

3. **Efficient Loop Conditions:**
   − The fast code uses 'i < 10' instead of 'i <= 9', which is slightly more efficient as comparing against 10 might be faster, though this is a minor optimization.

4. **Reduced Whitespace and Improved Code Structure:**
   − While not affecting runtime, the fast code has cleaner formatting, enhancing readability without impacting performance.

**JSON Output:**

```
{
''optimization_points'': [
  {
     ''description'': ''The slow code uses 'cout' which is slower due to object overhead. The fast code
     switches to 'printf', improving I/O efficiency.'',
     ''runtime_improvement'': 8,
     ''category'': ''Algorithm''
  },
  {
     ''description'': ''Precomputing the multiplication result ('mt = i ∗ j') before printing reduces
     redundant calculations in each loop iteration.'',
     ''runtime_improvement'': 6,
     ''category'': ''Algorithm''
  },
  {
     ''description'': ''Using 'i < 10' instead of 'i <=9' slightly improves loop condition efficiency, though
     the impact is minor.'',
     ''runtime_improvement'': 3,
     ''category'': ''Code Execution''
  }
]
}
```

Figure 5: Example of a runtime optimization instruction.

---

### A.2.1 INEFFICIENT ALGORITHMS

This category targets computational patterns such as recursion without memoization or arithmetic operations replaceable by bitwise alternatives. Such patterns are detected via function call tracking and control-flow analysis involving arithmetic operations and operand usage.

Figure 6 shows an example of directives generated by the symbolic advisor and the resulting optimized code. When given the raw recursive `fib` function, the detector matches it with the *recursion rule* from the Inefficient Algorithm category 1, which identifies recursion without memoization. Based on this match, it emits a directive template that recommends replacing recursion with memoization or dynamic programming. Guided by this directive, the downstream code-LLM produces the optimized version (B), where a `dp` buffer is introduced to eliminate redundant calls and improve runtime efficiency.

| (A) Input Code | (B) Optimized Code |
|---|---|
| ```c
int fib(int n){
    if (n <= 1) return n;
    return fib(n-1) + fib(n-2);
}
``` | ```c
int fib(int n){
    if (n <= 1) return n;
    if (dp[n] != -1)
        return dp[n];
    dp[n] = fib(n-1) + fib(n-2);
    return dp[n]; }
``` |

**(C) Bottleneck Diagnosis for recursion without memoization**

The following methods are purely recursive: [method: `fib`, lines: 1–4]. Applying memoization or dynamic programming can significantly reduce its execution time.

Figure 6: Memoization directive example. Left: slow recursive Fibonacci. Right: memoized version. Bottom: directive emitted by the symbolic advisor.

### A.2.2 SUBOPTIMAL DATA STRUCTURE USAGE

This category addresses inefficient use of data structures, such as employing dynamic containers (e.g., `std::vector`) when their dynamic capabilities are not utilized, or using non-hash maps where hash-based structures would provide better performance. Such inefficiencies are detected through type analysis and usage pattern tracking, which identify opportunities to substitute simpler or more efficient data structures.

Figure 7 illustrates an example where a dynamic vector is unnecessarily used to store values of predetermined size. The symbolic advisor detects that the vector does not leverage any dynamic behavior (e.g., resizing, insertion at arbitrary positions) and matches it with the static replacement rule. Based on this detection, it generates a directive template recommending replacement with a fixed-size array. Guided by this directive, the downstream code-LLM produces the optimized version (B), where the vector is replaced by a static array, eliminating allocation overhead and improving runtime efficiency.

| (A) Input Code | (B) Optimized Code |
|---|---|
| ```c
int n; scanf("%d",&n);
std::vector<int> v;
for(int i=0;i<n;++i) {
  v.push_back(i*i);
}
``` | ```c
int n; scanf("%d",&n);
int v[n];
for(int i=0;i<n;++i) {
  v[i] = i*i;
}
``` |

**(C) Bottleneck Diagnosis for vector without dynamic behavior**

The following vectors do not use dynamic operations: [variable: `v`, lines: 2–4]. Replacing them with a static array or fixed-size container can improve performance.

Figure 7: Static array directive example. Left: inefficient use of a dynamic vector. Right: optimized version using a static array. Bottom: directive emitted by the symbolic advisor.

### A.2.3 INEFFICIENT LIBRARY USAGE

This category addresses inefficient use of library functions, such as slow I/O operations (`cin`, `cout`) or expensive math functions (e.g., `pow`), which can often be replaced with faster alternatives. Such inefficiencies are detected by analyzing library call sites and inspecting the types and properties of their arguments.

Figure 8 presents an example where inefficient I/O operations are detected. The symbolic advisor identifies the use of `cin` and `cout`, which are known to incur higher overhead, and matches them with the *I/O replacement rule*. Based on this match, it generates a prompt recommending substitution with faster alternatives (`scanf` and `printf`). Guided by this prompt, the downstream code-LLM produces the optimized version (B), where the I/O calls are replaced, leading to improved runtime performance.

| (A) Input Code | (B) Optimized Code |
|---|---|
| ```int x, y;``` ```cin >> x >> y;``` ```int res = gcd(x, y);``` ```cout << res << endl;``` | ```int x, y;``` ```scanf("%d %d", &x, &y);``` ```int res = gcd(x, y);``` ```printf("%dn", res);``` |

**(C) Bottleneck Diagnosis for inefficient I/O library usage**

The following I/O library calls rely on slow operations: [call: `cin`, lines: 2–2, call: `cout`, lines: 4–4]. Replacing them with faster alternatives (`scanf`, `printf`) can improve performance.

Figure 8: I/O replacement directive example. **Left:** inefficient use of `cin`/`cout`. **Right:** optimized version with `scanf`/`printf`. **Bottom:** directive emitted by the symbolic advisor.

### A.2.4 INEFFICIENT LOOP USAGE

This category targets costly operations that are repeatedly executed inside loops but can be safely moved outside, such as sorting or redundant calculations. Such inefficiencies are detected by analyzing loop bodies and extracting loop-invariant computations.

Figure 9 illustrates an example where redundant operations are placed inside a loop. The symbolic advisor detects that both the sorting operation and the minimum-value extraction are loop-invariant, and matches this case with the *loop-invariant rule*. Based on this match, it generates a performance-aware prompt recommending that these computations be hoisted outside the loop. Guided by this directive, the downstream code-LLM produces the optimized version (B), where sorting and minimum extraction occur once before the loop, eliminating redundant work and improving runtime efficiency.

| (A) Input Code | (B) Optimized Code |
|---|---|
| ```int res = 0;``` ```for(int i=0;i<q;++i) {``` ```  std::sort(a, a + n);``` ```  int min_v = a[0];``` ```  res += min_v;``` ```}``` | ```int res = 0;``` ```std::sort(a, a + n);``` ```int min_v = a[0];``` ```for(int i=0;i<q;++i) {``` ```  res += min_v;``` ```}``` |

**(C) Bottleneck Diagnosis for redundant calls in loop**

The following redundant calls are placed inside loops: [call: `sort`, lines: 3–4]. Moving these calls outside the loop, or caching their results, can eliminate redundant work and improve efficiency.

Figure 9: Loop-invariant directive example. **Left:** redundant calls inside the loop. **Right:** optimized version with invariant computations moved outside. **Bottom:** directive emitted by the symbolic advisor.

## A.3 ROI RETRIEVER

The ROI retriever operates on the constructed ROI database to retrieve performance-relevant triplets. It first extracts a performance-related description of the given input code, similar to the ROI distillation process. At inference time, the inference model is prompted with a structured query asking it to describe the performance characteristics of the input code, as shown below.

---

**Prompt template for performance-related characteristic distillation from the given input code**

```
You are a competitive-programming performance analyst.

### Task
1. A **slow C++ program** is given between '''cpp''' blocks below.
2. Analyse it **only from a runtime-performance standpoint** - do **NOT
** propose fixes or rewrites.
3. Identify every major **bottleneck** that contributes to slower
runtime.
4. Cover these angles where applicable:
   * algorithmic complexity
   * data-structure choiceb
   * I/O or library usage
   * memory-access patterns / allocations
5. For each bottleneck, estimate its relative impact on a **1-10 scale
** (10 = largest slowdown factor).
```

---

For similarity measurement between the stored ROIs and the input analysis, we employ the HuggingFace model *Qodo/Qodo-Embed-1-1.5B*. Unlike the larger inference model, this lightweight embedding model can be run locally without burden, making it practical for retrieval. Moreover, it has been jointly trained on both natural language and code, making it well-suited to handle our setting where we compare code itself together with its natural-language description.

**(A) Input Code**

```cpp
int k; string s;
cin >> k >> s;
if(s.length() > k)
{
    for(int i=0;i<k;i++)
        cout << s[i];
    cout << "...";
}
else
    cout << s;
```

**(B) Retrieved Code Pair**

```cpp
\\------Slow Code------
int main(){
    string s;
    getline(cin,s);
    if((s.front()==s.back())
        ^ (s.length() % 2))
        cout << "Case 1" << endl;
    else
        cout << "Case 2" << endl;
}
```

```cpp
\\------Fast Code------
char s[100005];
int main() {
    int l = 0;
    for(char c=getchar();c!='\n'
        ;ch=getchar(), l++){
        s[l] = ch;
    }
    if ((s[0]==s[l-1]) ^ (l%2))
        printf("Case 1");
    else
        printf("Case 2");
}
```

**(C) Runtime Optimization Instruction**

1. Input Method: The slow code uses `cin >> s`, which is slower due to C++ stream overhead. The fast code replaces with direct `getchar()` calls, ...
2. String Handling: The slow code uses `std::string`, which adds memory and function call overhead, unlike the fixed-size array in the fast code.
3. Output Method: Replacing `cout` with `printf` in the fast code results in faster output operations.

Figure 10: Illustration of (A) the original input code, (B) the retrieved slow–fast code pair selected by the ROI retriever, and (C) the corresponding performance-related optimization instruction.

We analyze what our ROI retriever returns when given the random (A) input code in Figure 10. The given input simply prints the first k characters of s, appending "..." when the string is truncated. Although (B) the retrieved code pair performs a *different* task from (A), its selection is driven by a similarity of the performance aspects. Consequently, (C) optimization instruction extracted from (B) the retrieved slow code precisely identify its bottlenecks and also pinpoints the bottlenecks of (A): replacing slow `cin`/`cout` I/O with faster C-style functions, and avoiding the overhead associated with `std::string` by using fixed-size buffers. In contrast, a plain RAG instead chooses a snippet

of function that generates a string under certain conditions, a task superficially similar to (A). Despite its high code-to-code embedding score, it provides no insight into handling string operations or I/O overhead. Thus, despite surface syntax differences, our ROI retriever effectively captures the key performance themes relevant to the input code.

# B    IMPLEMENTATION DETAILS

All approaches considered in this work, including ECO, guide code-LLMs purely through prompting, except for Supersonic and fine-tuning–based methods. For these prompting-based baselines, we employ the *Qwen2.5-Coder* family as the inference model, executed via the ollama framework. In our setup, ollama runs models in a quantized configuration, specifically Q4_K_M, which is a 4-bit quantization scheme with grouped quantization designed to balance memory reduction and inference efficiency. We adopt this setting throughout our experiments. The decoding temperature is fixed at 0.7. The maximum input length is set to 4,096 tokens; if the source program exceeds this limit, it is truncated to fit within the context window. The maximum output length is set to 8,192 tokens to ensure that optimized programs and associated reasoning can be fully generated. For closed-source inference, we additionally evaluate *GPT-4o-mini* and *GPT-o4-mini*, accessed via their official API with default decoding parameters.

## B.1    PROMPT FORMAT OF ECO

Given an input code, the symbolic advisor applies its rules to detect performance bottlenecks. For each identified bottleneck, it generates a description specifying where and how the code should be optimized. These descriptions are then instantiated into the prompt template (Fig. 11), providing the model with explicit optimization guidance in a structured format.

```
Given a program and optimization tips, optimize the program and provide
 a more efficient version.

### Explanation:
1. {where_and_how_to_optimize1}
2. {where_and_how_to_optimize2}
...

### Original code:
{src_code}

### Optimized Code:
```

Figure 11: Prompt template for symbolic advisor.

The instruction retriever operates analogously to other retrieval-based baselines such as ICL, RAG, and SBLLM, supplying 2-shot examples in the prompt. It shares the same prompt template as these baselines, with one key distinction: in addition to the slow–fast code pairs, it also supplies the corresponding optimization instructions (Fig. 13).

We can combine these performance-aware prompts into a single prompt by slightly modifying and concatenating the two completed templates. The source code (src_code) needs to be included only once, at the end.

## B.2 PROMPT FORMAT OF GENERIC PROMPTING BASELINES

Instruction-only and CoT are adopted from the prompting methods used in PIE (Shypula et al., 2024). The instruction-only setting (Fig. 12) simply provides the input code without any external context and requests an optimized version. In contrast, CoT augments the prompt by prepending a system message enclosed in square brackets, explicitly instructing the model to output its reasoning process.

Methods that rely on retrieval, such as ICL, RAG, and SBLLM, use two slow–fast code pairs and share the same prompt template as ECO's Instruction Retriever. The only difference is that they exclude the optimization instructions enclosed in square brackets (Fig. 13).

```
[You are a software developer and now you will help to improve code
efficiency. Explain the reasons briefly at the beginning.]

Optimize the program and provide a more efficient version.

### Original Code:
{src_code}

### Optimized Code:
```

Figure 12: Prompt template for Instruction-only and CoT.

```
Optimize the program and provide a more efficient version. Followings
are retrieved examples for optimization.

### Original Example Code1:
```{slow_code1}```

### Optimized Example Code1:
```{fast_code1}```

[{optimization_instruction1}]

### Original Example Code2:
```{slow_code1}```

### Optimized Example Code2:
```{fast_code2}```

[{optimization_instruction2}]

Now, optimize the following code.

### Original Code:
{src_code}

### Optimized Code:
```

Figure 13: Prompt template for retrieval methods.

## B.3 Detail Implementation Setting

### B.3.1 Implementation of Supersonic

We closely follow the official implementation of Supersonic[2]. In their public repository, the authors release a trained CodeBERT-based encoder–decoder model, configured with beam search (num_beams=10). The model targets C++ and is trained on multiple datasets from CodeNet—including AIZU and AtCoder—of which PIE is a subset, ensuring a similar distribution. Importantly, Supersonic is trained not to output the fast implementation directly but rather to generate the diff patch between the slow and fast code. We adopt this framework as-is and evaluate it on our PIE test set: given a slow program, the shared model generates a patch, which we then apply to reconstruct the final optimized code. This evaluation therefore corresponds to an in-distribution setting.

### B.3.2 Implementation of SBLLM

We closely follow the official implementation of SBLLM[3]. SBLLM extends the RAG framework into an iterative scheme, where in each iteration it performs execution-based candidate selection, ranks the candidates, and updates the code by prompting the LLM again with the top-ranked examples retrieved by RAG.

For a fair comparison, we align the model and environment settings of SBLLM with those of ECO's inference setup. The only difference is that SBLLM initializes the first candidate code using CoT prompting, for which we directly adopt the official SBLLM prompt template. As in our main experiments, RAG retrieval is performed on the PIE HQ dataset and inference uses *Qwen2.5-coder:7B*. For SBLLM-specific hyperparameters, we follow the defaults in the official code, setting the number of selected representative samples to 3 and the maximum iteration number to 4.

### B.3.3 Implementation of Fine-tuning

We additionally conducted experiments with the fine-tuning approach, which are reported in Appendix D.4. For reproducibility, we closely follow the official implementation of PIE[4]. Since PIE also uses our reference code-pair corpus (PIE HQ), we fine-tune the *Qwen2.5-Coder-7B* model[5] and *GPT-4o-mini*[6] on the same dataset.

For *Qwen2.5-Coder-7B*, we use key hyperparameters of batch size 32 (micro-batch size 2), learning rate $1 \times 10^{-5}$, and cutoff length 2000, and employ early stopping until convergence (approximately four epochs). Following the original configuration, training is performed using the HuggingFace Transformers library with FSDP enabled, distributed across 8×48GB NVIDIA RTX A6000 GPUs. For *GPT-4o-mini*, we use the configuration provided by the model provider, with batch size 8, learning rate multiplier 1.8, and train for 3 epochs until convergence.

### B.3.4 Implementation of Runtime Reasurements

We employ the gem5 system simulator to obtain reliable performance measurements. Gem5 provides cycle-accurate emulation of modern microarchitectures, allowing deterministic programs to yield deterministic runtime results. This property ensures reproducibility in research and reduces measurement noise. Specifically, we adopt the Verbatim configuration of Intel Skylake from gem5[7] , which also allows our framework to be extended to other platforms such as ARM or RISC-V without requiring direct hardware access.

In contrast to lightweight profilers such as Hyperfine sharkdp (2023), which are faster but prone to high variance due to system noise, gem5 offers consistent and denoised performance measurements at the cost of higher computational overhead. This motivated our dataset refinement step to reduce redundancy.

---

[2] https://github.com/ASSERT-KTH/Supersonic

[3] https://github.com/shuzhenggao/sbllm

[4] https://github.com/LearningOpt/pie

[5] https://huggingface.co/Qwen/Qwen2.5-Coder-7B

[6] Version: gpt-4o-mini-2024-07-18

[7] https://hub.docker.com/r/alexshypula/gem5-skylake:api

# C DATASET DETAILS

We utilize datasets that provide both source codes and corresponding test cases in code optimization tasks. The goal is to modify a given input code so that it runs faster while preserving its original functionality, which is verified through test cases. In this work, we focus on the C++ language.

The training dataset is used either for direct fine-tuning or for retrieval, and serves as the knowledge base of known optimization patterns. For this purpose, we adopt the HQ dataset, a high-quality subset of the PIE training data. The original PIE dataset consists of C++ slow–fast code pairs, submitted by human programmers on coding problems from CodeNet. There can be multiple pairs of solution for one coding problem. The HQ subset is pruned to retain pairs with the highest speedup while limiting each problem to at most four submissions, alleviating data imbalance issues.

For evaluation, we require datasets that provide input codes along with test cases. Unlike the training data, only the input (slow) code is necessary; optimization is expected to be performed during evaluation. We consider three complementary test sets:

- (1) PIE test set: shares similar characteristics with the HQ dataset but covers disjoint CodeNet problems, and
- (2) Codeforces dataset: introduces a more challenging out-of-distribution setting.
- (3) HumanEval-X dataset: provides canonical reference implementations originally designed for code-generation tasks.

The PIE test set contains 255 samples, the Codeforces test set consists of 300 samples, and each HumanEval-X split contains 164 samples. All problems across the three datasets are accompanied by curated test cases. Together, these datasets enable a comprehensive evaluation covering in-distribution, out-of-distribution, and code-generation–oriented scenarios.

Table 5: Statistics of datasets used in our study. We report the number of code pairs, problems, and per-problem statistics. *Org.* refers to the original test set before applying our curation procedure.

| Dataset | # Samples | # Problems | Max Samples / Problem | Avg. Samples / Problem |
|---|---|---|---|---|
| PIE-Cpp-Train | 77,967 | 1,474 | 670 | 14.73 |
| PIE-Cpp-HQ | 4,085 | 1,474 | 4 | 2.77 |
| PIE-Python-Train | 36,857 | 1,644 | 276 | 22.42 |
| PIE-Python-HQ | 5,307 | 1,644 | 4 | 3.23 |
| PIE-Cpp-Test (Org.) | 978 | 41 | 481 | 23.85 |
| PIE-Cpp-Test | 255 | 41 | 100 | 1.00 |
| PIE-Python-Test (Org.) | 1,000 | 213 | 46 | 4.69 |
| PIE-Python-Test | 100 | 100 | 4 | 1.64 |
| Codeforces-Test | 300 | 30 | 10 | 10.00 |
| HumanEvalX-Cpp-Test | 164 | 164 | 1 | 1.00 |
| HumanEvalX-Python-Test | 164 | 164 | 1 | 1.00 |

## C.1 PIE TESTSET

The original PIE test set consists 978 pairs drawn from 41 coding problems. We identify two major issues that hinder fair and efficient evaluation: (1) severe problem imbalance—over 600 pairs origgiante from just 3 problems; and (2) redundancy in test cases— the official 104 test cases include many exact or near-duplicate overlaps. Redundancy is especially problematic while we profile with use gem5 to obtain stable runtime measurements, a process that can take multiple days in this settings. Notably, the PIE authors also caution that practitioners may need to tailor a lighter-weight evaluation to their setting.

We (i) rebalance the problem distribution by capping the number of examples per problem to at most 10, and (ii) remove redundant test cases by clustering candidate inputs using $n$-gram similarity and discarding exact or highly similar duplicates. When more than 10 cases remain after de-duplication, we prioritize official (public/private) cases over LLM-generated cases and retain up to 10 per problem. Our curated PIE test set comprises 255 evaluation instances, each accompanied by at most 10 carefully selected, non-redundant test cases. This refinement mitigates imbalance while preserving task diversity, and substantially reduces the computational burden of `gem5`-based profiling.

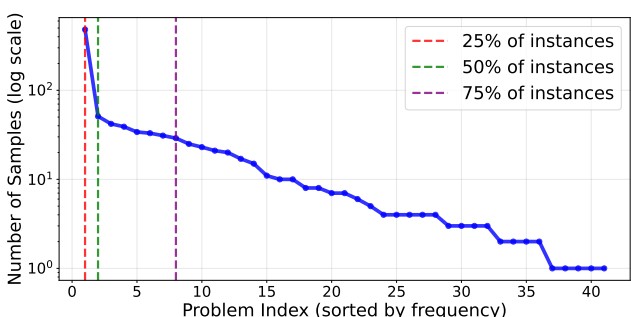

Figure 14: Long-tail data distribution of original PIE-Cpp-Test.

## C.2 CODEFORCE TESTSET

We construct an additional out-of-distribution benchmark from the Codeforce dataset. First we extract Codeforce data from `deepmind/code_contests`, other source can be contaminated or duplicated with CodeNet. To ensure consistency with our evaluation setting, we apply several filtering steps: (i) retain only C++ solutions, (ii) require at least 10 available test cases (from public or private sets, and (iii) discard problems with a time limit greater than 2 seconds. For each selected problem, we uniformly sample up to 10 test cases, prioritizing public cases when available, and generate corresponding input–output files. We further sample up to 10 candidate solutions per problem to form evaluation instances. Finally, we select 30 problems meeting these criteria, resulting in 300 code–test pairs accompanied by curated test cases. This procedure yields a balanced and computationally feasible Codeforces test set while preserving the diversity of problem domains and difficulty levels.

## C.3 HUMANEVALX TESTSET

To further evaluate ECO under a more challenging out-of-distribution setting, we additionally construct a benchmark based on a canonical code generation dataset where no slow–fast code pairs are provided. For this purpose, we use the HumanEval-X dataset[8] , which shares the same problem set as the widely used HumanEval benchmark while providing canonical C++ and Python solutions.

HumanEval-X consists of 164 samples. Each sample includes: (i) a natural-language problem *prompt* describing the target functionality; (ii) a *declaration* containing includes and the required function signature; (iii) a *test* snippet implemented as inline assertion-based checks; and (iv) a reference *canonical solution*.

This dataset differs from our slow–fast pair setting in several ways. First, the task is function-level code generation from natural-language descriptions, and no explicit slow–fast code pairs are supplied. The canonical solution is not necessarily a slow implementation; nevertheless, we treat it as the "slow" code that ECO aims to optimize. Second, the test cases are not given as input–output pairs but as inline assertions executed directly on the candidate implementation.

Following this structure, we construct the HumanEval-X test set by concatenating each sample's *declaration* and *canonical solution*, treating the result as the slow code to be optimized. All 164 samples are included, resulting in a comprehensive assertion-based evaluation set for measuring ECO's optimization performance in a pure code-generation context.

---

[8]https://huggingface.co/datasets/zai-org/humaneval-x

## C.4 PIE-PYTHON HQ AND TESTSET

To evaluate the cross-language scalability of ECO beyond C++, we additionally measure runtime improvement on Python programs. For this purpose, we use the PIE-Python dataset[9] , provided alongside the PIE study Shypula et al. (2024). Similar to PIE-Cpp, PIE-Python consists of Python submissions sourced from the CodeNet dataset, and it shares the same set of problems and test cases as PIE-Cpp.

PIE-Python provides train, validation, and test splits, but does not include a separate high-quality (HQ) training set. Following the procedure used in the PIE study Shypula et al. (2024), we construct an HQ subset by selecting, for each problem, the slow–fast submission pair with the highest speedup while disallowing more than four such pairs per problem. This results in a total of 5,307 Python-HQ examples, which we use as the retrieval corpus for ECO.

For PIE-Python-Test, we apply the same data curation strategy used in our main evaluation, allowing up to four submissions per problem. From this curated pool, we randomly sample 100 instances to form the final evaluation set. This supplementary benchmark enables us to assess ECO's effectiveness on a different programming language while maintaining full compatibility with the PIE evaluation protocol.

---

[9]`https://github.com/madaan/pie-perf`

# D ADDITIONAL EXPERIMENTS & ANALYSIS

## D.1 DETAILED ANALYSIS OF MAIN EXPERIMENT

We provide additional analysis of the results in Fig. 2. As expected, instruction-only prompting yields relatively low performance since it represents the default baseline. Interestingly, however, CoT underperforms even compared to instruction-only. This degradation mainly stems from verbose debug outputs (e.g., `cout << ''time:''`) that increase runtime overhead. These findings suggest that CoT prompting can be redundant for models such as *Qwen-Coder*, which are already pretrained with multi-step reasoning capabilities.

Surprisingly, optimization-oriented baselines such as Supersonic and SBLLM also underperform relative to generic prompting methods, largely due to their lower correctness as noted in the main text. Even in the SBLLM paper Gao et al. (2025), for example, SBLLM achieved only a $1.22\times$ speedup under the Best@5 on the same PIE-Cpp test set (using GPT), despite targeting a similar distribution. Although our dataset was pruned, the distributional characteristics remain consistent.

Supersonic's reported performance in its paper (Chen et al., 2024) is not directly comparable since the datasets differ, but it is worth noting that its speedups were lower than those of GPT-3.5-Turbo and GPT-4 under standard instruction prompting. In contrast, our ECO framework combined with *Qwen2.5-Coder:7B* significantly outperforms instruction-only prompting on both *GPT-4o-mini* and *GPT-o4-mini*.

Overall, these results highlight the difficulty of achieving meaningful optimization gains, and demonstrate that ECO is able to unlock substantial improvements in runtime efficiency using prompting alone.

## D.2 EXPERIMENT FOR LANGUAGE SCALABILITY: PIE-PYTHON

We conduct additional experiments on Python programs to evaluate whether ECO generalizes beyond C++ and exhibits cross-language scalability. Using the PIE-Python dataset constructed in Appendix C.4, we employ the HQ subset for ROI distillation and retrieval, and use 100 test samples for evaluation. For this experiment, we focus on the ROI Retrieval–only variant of ECO. The symbolic advisor requires language-specific Joern query rules, which need some manual effort. We leave language-specific symbolic reasoning to future work and instead evaluate the generality and portability of the ROI retrieval mechanism.

Table 6: Average performance of the baseline and the ROI-Retriever–only variant of ECO on the PIE-Python dataset over 10 trials.

| Model | Methods | Best@1 | | | Best@5 | | |
|---|---|---|---|---|---|---|---|
| | | ACC(%) | SP | OPT(%) | ACC(%) | SP | OPT(%) |
| Qwen2.5-coder:3b | Instruction-only | 17.80 | 1.09× | 1.70 | 43.70 | 1.41× | 7.40 |
| | ECO w/o SA | 17.50 | 1.18× | 2.40 | 44.80 | 1.67× | 9.50 |
| Qwen2.5-coder:7b | Instruction-only | 30.20 | 1.41× | 6.80 | 59.60 | 2.21× | 18.60 |
| | ECO w/o SA | 31.70 | 1.44× | 6.30 | 64.40 | 2.22× | 19.00 |
| Qwen2.5-coder:14b | Instruction-only | 37.60 | **1.82×** | 9.50 | 59.30 | 2.47× | 19.30 |
| | ECO w/o SA | **44.95** | 1.81× | **12.58** | **68.87** | **2.72×** | **27.53** |

The results in Table 6 summarize ECO's performance on Python. While the improvements are modest for smaller models or single-trial evaluations, the gains become more pronounced as model size and the number of trials increase. In particular, for the 14B model under the Best@5 setting, ECO achieves substantial improvements across ACC, SP, and OPT. This trend is consistent with our main findings: the ROI Retriever excels at exploring a broader optimization search space and generating diverse candidate optimizations, which is especially beneficial when multiple attempts are allowed. Furthermore, as model capacity increases, the LLM follows ECO's guidance more faithfully, resulting in larger performance improvements. These results demonstrate that ECO's ROI retrieval component can be effectively transferred across languages with minimal additional effort, yielding meaningful runtime improvements even without language-specific symbolic analysis.

## D.3 EXPERIMENT FOR NON-OPTIMIZATION BENCHMARK: HUMANEVAL-X

In previous experiments, we observed substantial performance improvements on datasets specifically designed for runtime optimization—such as PIE-Cpp, PIE-Python, and Codeforces—where each problem contains multiple slow–fast code pairs. We seek to examine whether a code optimization task can also be evaluated on benchmarks constructed for entirely different purposes. To this end, we additionally evaluate ECO on the HumanEvalX dataset (Appendix C.3), which is originally designed for code generation from natural-language descriptions rather than optimization.

Table 7: Performance of the baseline and ECO on the HumanEvalX-Cpp and HumanEvalX-Python datasets.

| Dataset / Model | Method | Best@1 | | | Best@5 | | |
|---|---|---|---|---|---|---|---|
| | | ACC(%) | SP(%) | OPT(%) | ACC(%) | SP(%) | OPT(%) |
| **HumanEvalX-Cpp** | | | | | | | |
| Qwen2.5-coder:3b | Instruction-only | 37.20 | 0.01 | 0.00 | 71.95 | 5.10 | 3.05 |
| | ECO | 38.41 | 4.51 | 2.44 | 73.78 | 9.25 | 7.32 |
| Qwen2.5-coder:7b | Instruction-only | 46.34 | 8.94 | 1.83 | 79.88 | 21.91 | 5.49 |
| | ECO | 36.59 | 26.03 | 6.71 | 76.83 | 62.04 | 17.07 |
| Qwen2.5-coder:14b | Instruction-only | 67.68 | 0.29 | 1.83 | 87.20 | 0.36 | 3.05 |
| | ECO | 55.49 | 21.42 | 7.32 | 88.41 | 49.11 | 14.63 |
| **HumanEvalX-Python** | | | | | | | |
| Qwen2.5-coder:3b | Instruction-only | 67.07 | 0.18 | 0.00 | 92.07 | 0.73 | 0.61 |
| | ECO w/o SA | 55.49 | 0.85 | 2.44 | 91.46 | 2.42 | 7.93 |
| Qwen2.5-coder:7b | Instruction-only | 72.56 | 0.62 | 0.61 | 95.73 | 1.50 | 3.05 |
| | ECO w/o SA | 73.17 | 1.07 | 1.22 | 95.12 | 4.02 | 7.32 |
| Qwen2.5-coder:14b | Instruction-only | 78.66 | 0.22 | 0.00 | 96.34 | 0.95 | 0.61 |
| | ECO w/o SA | 76.40 | 0.85 | 0.62 | 95.03 | 4.59 | 6.83 |

Table 7 reports the performance of ECO and the baseline across both HumanEvalX-Cpp and HumanEvalX-Python. ECO consistently outperforms the instruction-only baseline in ACC, SP, and OPT. For instance, on HumanEvalX-Cpp, ECO achieves substantially higher speedups—such as 26.03% versus 8.94% under Best@1 for QwenCoder-7B, and 62.04% versus 21.91% under Best@5—demonstrating that ECO can provide useful optimization guidance even in non-optimization-oriented benchmarks. The Python results show similar trends but with smaller magnitudes; this is because (i) the Python evaluation uses the ROI Retriever-only variant of ECO, and (ii) Python programs generally present smaller optimization margins due to their high-level abstractions.

At the same time, the overall improvements on HumanEvalX remain limited compared to the substantial gains observed on PIE and Codeforces. A key factor is that HumanEvalX provides a single canonical solution per problem, which we treat as the "slow" implementation. These canonical solutions are often already highly optimized. Figure 15 shows representative examples: constant-time summation using the Gaussian formula, and the Euclidean algorithm for GCD—both of which are near-optimal in practice. Because such solutions leave little algorithmic headroom, even large models such as QwenCoder-14B cannot achieve large speedups, and ECO's improvements saturate accordingly. As noted earlier, an SP of, for example, 21.91% corresponds to only a 1.22× speedup, highlighting the narrow room for further acceleration. In summary, ECO remains effective on HumanEvalX, but the improvements are naturally constrained by the near-optimal nature of the canonical solutions.

```
P13: greatest_common_divisor

int greatest_common_divisor(int a, int b){
    int out, m;
    while (true){
        if (a < b) {
            m = a; a = b; b = m;
        }
        a = a % b;
        if (a == 0) return b;
    }
}
```

```
P24: largest_divisor

int largest_divisor(int n){
    for (int i = 2; i * i <= n; i++)
        if (n % i == 0) return n / i;
    return 1;
}
```

```
P60: sum_to_n

int sum_to_n(int n){
    return n*(n+1)/2;
}
```

```
P70: strange_sort_list

vector<int> strange_sort_list(vector<int> lst){
    vector<int> out = {};
    sort(lst.begin(), lst.end());
    int l = 0, r = lst.size() - 1;
    while (l < r){
        out.push_back(lst[l]);
        l += 1;
        out.push_back(lst[r]);
        r -= 1;
    }
    if (l == r) out.push_back(lst[l]);
    return out;
}
```

Figure 15: Four canonical example from HumanEval-X.

### D.4 INDIRECT COMPARISON WITH FINE-TUNE METHOD

We do not directly compare ECO with fine-tuned models, as prior works similarly argued that such comparisons are not entirely fair (Shypula et al., 2024; Gao et al., 2025; Chen et al., 2024). We discuss fine-tuning separately for two main reasons. First, its applicability is limited: it cannot be applied to some recent closed-source models such as GPT-5, preventing direct use with state-of-the-art systems. Second, it requires substantial GPU resources and training time, making it impractical for rapid or large-scale deployment. Nevertheless, for reference, we conducted additional fine-tuning experiments.

Table 8: Performance of the fine-tuning baseline and ECO across *Qwen2.5-coder:7b* and *GPT-4o-mini*, representing open-source and closed-source models, respectively.

| Model | Methods | Best@1 | | | Best@5 | | |
|---|---|---|---|---|---|---|---|
| | | ACC(%) | SP | OPT(%) | ACC(%) | SP | OPT(%) |
| Qwen2.5-coder:7b | Fine-tune | 46.63 | 2.23× | 25.28 | 79.65 | 3.73× | 52.12 |
| | ECO | 36.27 | 2.15× | 23.84 | 74.24 | 3.26× | 48.04 |
| GPT-4o-mini | Fine-tune | 42.90 | **2.71×** | 29.88 | 75.29 | **4.54×** | 54.90 |
| | ECO | **74.90** | 2.58× | **34.51** | **89.51** | 3.97× | **60.78** |

Fine-tuning is performed until convergence on the PIE HQ dataset, following the setup in Appendix B.3.3. As shown in Table 8, the open-source model *Qwen2.5-Coder-7B* achieves a 3.73× speedup after fine-tuning—slightly higher than the 3.26× obtained using ECO on the same model. For the stronger closed-source model *GPT-4o-mini*, however, ECO markedly outperforms fine-tuning in ACC, with more than a 30% improvement in the Best@1 setting, and also achieves higher OPT performance. Fine-tuning yields a marginally higher SP score.

These results highlight two key insights. First, higher-capacity LLMs more faithfully follow ECO's optimization guidelines, allowing ECO to benefit directly from model scaling—unlike fine-tuning, which primarily captures pattern-level correlations and thus exhibits limited gains as model size increases. This trend aligns with the general cross-model patterns observed in Table 4. Second, ECO shows clear strengths on ACC and OPT rather than SP, suggesting that its targeted and robust guidance provides more reliable signals than what fine-tuning can internalize. In other words, ECO reduces the risk of poor outputs while improving actionable optimization quality.

Overall, while fine-tuning can offer modest improvements under narrow and model-specific conditions, ECO provides a more practical, stable, and broadly scalable solution across diverse models and deployment settings.

### D.5 SYMBOLIC ADVISOR DIRECTIVE QUALITY

We evaluate whether optimization methods can consistently resolve easy bottleneck cases by reapplying our slow I/O library usage detection rule——one of the most frequent and apparent bottleneck types——to all outputs, regardless of their functional correctness. We measure the proportion of outputs in which the previously identified slow I/O bottleneck is no longer detected after optimization.

Table 9: Accuracy and proportion of resolved I/O bottlenecks for different methods.

| Methods | ACC (%) | Resolved Bottleneck (%) |
|---|---|---|
| Instruction | 33.61 | 22.40 |
| RAG | 29.06 | 48.09 |
| Supersonic | 7.06 | **80.33** |
| ECO (S) | **48.59** | 78.14 |

As shown in Table 9, a substantial portion of these bottlenecks remains unresolved in the Instruction and RAG methods, which lack the explicit capability to pinpoint *where* optimizations are necessary. Interestingly, Supersonic, trained explicitly on slow–fast code pairs, shows some success in identify-

ing and removing inefficient patterns. However, it heavily fails to appropriately revise the removed code segments, thereby breaking code functionality with 7.06% ACC. This suggests that although data-driven learning can effectively highlight performance issues, it remains unreliable in revisions without explicit guidance for *how* to optimize.

### D.6   ROI DISTILLATION RELIABILITY

We conducted a evaluation of the ROI distillation to validate whether the distilled ROIs provide meaningful and helpful information. Evaluating the quality of extracted optimization knowledge in a deterministic or computationally verifiable manner is inherently challenging. Unlike code correctness, which can be verified through execution, the assessment of optimization analysis quality—such as whether the identified inefficiencies are truly the root cause or whether the prioritization accurately reflects performance impact—lacks objective ground truth. To address this limitation, we employ an LLM-as-a-judge approach to analyze the reliability of our ROI distillation process.

#### D.6.1   EVALUATION OF ROI IDENTIFICATION AND EXPLANATION QUALITY

We first assess whether ROIs correctly identify inefficiencies and provide accurate explanations. We generated two types of samples: (1) matching samples, where each code pair is correctly matched with its corresponding ROI, and (2) random matching samples, where code pairs are randomly paired with ROIs from different code pairs. We extracted 100 samples for each condition.

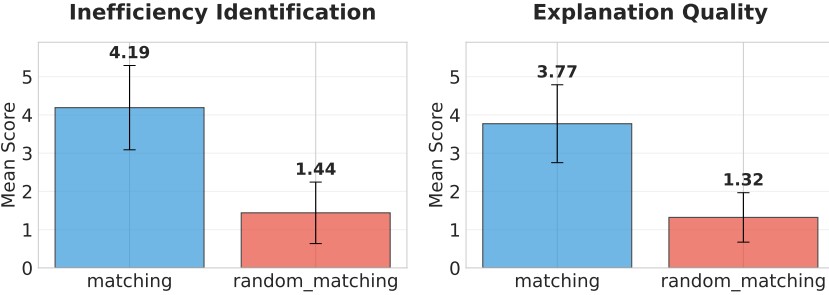

Figure 16: Average scores of Inefficiency Identification and Explanation Quality for the two types, demonstrating that ROI distillation produces reliable, well-aligned optimization guidance.

Each sample is evaluated by GPT-4o across two key dimensions: Inefficiency Identification (II) and Explanation Quality (EQ). Both dimensions are scored on a 1-5 scale. The evaluation criteria and prompt used for this assessment is provided in Figure 17. As shown in Figure 16, the distilled ROIs accurately capture the underlying optimization insights. This demonstrates that our ROI distillation procedure produces meaningful, code-pair-specific optimization instructions rather than generic or misaligned descriptions.

#### D.6.2   EVALUATION OF IMPACT PRIORITIZATION

We conducted a binary comparison experiment to assess whether ROIs correctly prioritize optimization points based on their actual performance impact. For each of 100 samples, we compared two versions of the same ROI: Version A (original ROI with original runtime improvement scores) and Version B (ROI with reversed runtime improvement scores, where each score is transformed as new score = 10 - old score). Both versions identify the same optimization points with identical descriptions, differing only in their prioritization scores. We leverage GPT-4o to determine which version better prioritizes the impact of the identified optimizations. The evaluation prompt for this binary comparison is provided in Figure 18.

The results show that Version A (original ROI) was chosen as the better prioritization in 99 out of 100 cases (99.0%), with an average confidence score of 4.98 ± 0.14 on a 1-5 scale. This strong preference for the original prioritization indicates that the distilled ROIs accurately reflect the relative importance of different optimization points, validating that the runtime_improvement scores effectively capture the actual performance impact hierarchy.

```
You are an expert code optimization evaluator. Your task is to evaluate
 the quality of Runtime Optimization Instructions (ROI) that describe
how a slow code was optimized to become fast code.

Slow Code:
```{slow_code}```

Fast Code:
```{fast_code}```

ROI (Runtime Optimization Instructions):
{roi_text}

Please evaluate the ROI across three progressive dimensions, each on a
1-5 scale. These dimensions build upon each other: first identifying
the problem, then explaining it, and finally prioritizing it.

### 1. Inefficiency Identification (II)
Question: Does the ROI correctly identify the actual performance
inefficiency in the slow code?

Evaluate whether the ROI identifies the real bottleneck or optimization
 opportunity that exists in the slow code when compared to the fast
code.

Key consideration: Does the identified inefficiency actually exist in
the slow code and contribute to its slowness?

### 2. Explanation Quality (EQ)
Question: Does the ROI properly explain the reason and process for why
the identified inefficiency causes performance issues and how the
optimization addresses it?

Evaluate whether the ROI provides a clear, accurate explanation of:
- Why this inefficiency causes performance problems
- How the fast code addresses or eliminates this inefficiency
- The mechanism or process by which the optimization works

Key consideration: Does the explanation accurately describe the
relationship between the slow code's inefficiency and the fast code's
optimization?

---

Provide your evaluation in the following JSON format:
{{
    "ii_score": <integer 1-5>,
    "eq_score": <integer 1-5>,
    "ii_reasoning": "<brief explanation for II score>",
    "eq_reasoning": "<brief explanation for EQ score>",
}}
```

Figure 17: Prompt template for assessing Inefficiency Identification and Explanation Quality.

```
You are an expert code optimization evaluator. Your task is to compare
two versions of Runtime Optimization Instructions (ROI) and determine
which one better prioritizes the impact of optimizations.

Slow Code:
'''{slow_code}'''

Fast Code:
'''{fast_code}'''

ROI Description:
{roi_description}

Version A (Original ROI):
'''json
{roi_json_a}
'''

Version B (Reversed Priority ROI):
'''json
{roi_json_b}
'''

## Task:
Compare Version A and Version B. Both versions identify the same
optimization points with the same descriptions, but they differ in
their 'runtime_improvement' scores (1-10 scale).

Question: Which version (A or B) better prioritizes the impact of the
identified optimizations? In other words, which version assigns higher
scores to optimizations that actually have greater impact on
performance improvement?

Consider:
- Which optimizations actually contribute more to the performance
difference between slow and fast code?
- Do the runtime_improvement scores in each version accurately reflect
the relative importance of each optimization?
- Which version's prioritization better matches the actual performance
impact?

Provide your answer in the following JSON format:
{{
    "better_version": "A" or "B",
    "reasoning": "<brief explanation of why this version better
    prioritizes impact>",
    "confidence": <integer 1-5, where 5 is very confident>
}}
```

Figure 18: Prompt template for assessing Impact Prioritization.

## D.7 ROI RETRIEVER PROMPT DESIGN VARIATIONS

In ECO, the ROI retriever facilitates optimization by supplying the model with the most relevant past ROIs and their corresponding slow–fast code pairs. Our default configuration follows a 2-shot prompting scheme and uses the *Full response* of the distilled ROI, which appears after the `</think>` marker (as described in Appendix A.1 and illustrated in Figure 5).

In this section, we explore whether alternative ways of presenting retrieved ROIs and their associated slow–fast code pairs can lead to more effective guidance during inference. This question is particularly relevant for smaller models, which may struggle to process the relatively verbose and long ROIs and code-pair contexts. Prompt simplification may therefore offer a potential remedy for improving correctness without sacrificing speedup. To examine the impact of such variations, we evaluate two alternative prompt designs in addition to the default setup, focusing our analysis on the smallest model, *Qwen2.5-Coder-3B*.

Table 10: Performance of Qwen2.5-coder:3b under different ROI Retriever prompting strategies on PIE-Cpp, using the ECO ROI-retrieval–only variant.

| Variation Strategy | Best@1 | | | Best@5 | | |
|---|---|---|---|---|---|---|
| | ACC(%) | SP | OPT(%) | ACC(%) | SP | OPT(%) |
| **Shot Count Variants** | | | | | | |
| 1-shot | 14.12 | 1.32× | 3.14 | 37.25 | 1.66× | 12.55 |
| 2-shot (Full Response, default) | 14.51 | 1.16× | 4.71 | 41.18 | 1.60× | 16.86 |
| 3-shot | 16.86 | 1.18× | 5.49 | 38.82 | 1.65× | 16.08 |
| **Prompt Format Variants** | | | | | | |
| Full Response (default) | 14.51 | 1.16× | 4.71 | 41.18 | 1.60× | 16.86 |
| Bullets | 18.04 | 1.17× | 5.10 | 40.39 | 1.67× | 13.73 |
| No Code Pair | 14.51 | 1.37× | 7.84 | 32.94 | 1.83× | 17.25 |

### D.7.1 EFFECT OF VARYING THE NUMBER OF RETRIEVED EXAMPLES

In this experiment, we analyze how varying the number of retrieved ROI–code-pair examples (1-shot, 2-shot, and 3-shot) affects optimization performance. Our default configuration uses 2-shot prompting, but we explore whether providing more retrieved examples leads to better optimization, or conversely, whether smaller models benefit from receiving only a single, highly relevant example due to their limited capacity.

As shown in Table 10, the overall performance of the three variants is largely similar, and no configuration consistently outperforms the others across all metrics. Notably, the 1-shot variant achieves the highest SP under Best@1, which aligns with the intuition that focusing on a single, highly relevant retrieved example can help the model commit to one optimization trajectory more effectively. However, its OPT scores are the lowest among the three variants, indicating that relying on just one example limits the model's ability to capture a broader range of optimization strategies.

We hypothesized that shorter prompts might benefit smaller models by reducing cognitive load. However, the overall results do not support this. This suggests that the primary bottleneck for small models is not prompt verbosity (length) but rather their limited capacity to interpret and execute the optimization intentions conveyed by the ROIs.

### D.7.2 EFFECT OF ROI SIMPLIFICATION AND CODE-PAIR REMOVAL

We investigate how each retrieved ROI–code-pair entry should be incorporated into the prompt. In the default setting, ECO provides the full distilled ROI (*Full response*) together with the corresponding slow–fast code pair. We derive two simplified variants of this configuration:

- *Bullets*: retains only the itemized optimization steps from the full response while omitting the narrative comparison, and still includes the code pair;

- *No-codepair*: preserves the full response but removes the example code pair entirely, leaving only textual optimization guidance.

These variants allow us to isolate the effects of (i) compressing ROI into shorter, structured hints and (ii) removing explicit code-level trajectories.

The results in Table 10 show that each variant has a distinct trade-off. The *Bullets* variant achieves the highest ACC under Best@1 (18.04% compared to 14.51%, while maintaining comparable performance under Best@5. This suggests that providing concise, itemized optimization hints helps the small model focus on a specific modification direction in a single attempt. In contrast, the No-codepair variant yields the highest SP and OPT values (e.g., SP@5 = 1.83$\times$ and OPT@5 = 17.25%), but at the cost of lower ACC, indicating that removing the concrete code trajectory encourages more aggressive but less reliable transformations.

### D.8 BEAM SEARCH AS AN EXTENSION TO ECO

In addition to single-step optimization, we also examine how ECO behaves when combined with a beam-search–style procedure, as in prior work on RAS for program optimization (**?**). RAS improves performance by iteratively generating multiple candidate optimizations, executing them, and recursively continuing the search from the best-performing candidate. In contrast, our main ECO setting focuses on single-round generation guided by retrieved ROIs and does not perform multi-step search or runtime-based candidate selection. These two dimensions—how to guide the model (ECO vs. contextual retrieval) and how to explore the search space (single step vs. beam search)— are orthogonal and can be combined.

We adapt the RAS beam-search procedure on top of ECO's ROI-Retriever–only variant to study this interaction under comparable conditions. Using *Qwen2.5-coder:7B* on the PIE-Cpp dataset with the same gem5 configuration, we run beam search for up to three iterations. At each iteration, ECO generates $k = 5$ candidate programs; among those that pass all test cases, we select the candidate with the highest speedup as the seed for the next iteration. Unlike our default Best@5 evaluation, where the retriever and analysis may vary across samples due to temperature, the beam-search experiments use a deterministic configuration: the same retrieved ROI–code-pair context is used for all candidates in an iteration, and once a retrieved pair has been used to produce a best-performing candidate, it is not reused in subsequent iterations. This setup allows a fair comparison between ECO's ROI-guided optimization and RAS-style beam search under the same model and benchmark.

Table 11: Performance across beam-search steps (Best@5 only).

| Step | ACC(%) | SP | OPT(%) |
|---|---|---|---|
| STEP 1 | 48.24 | 2.73$\times$ | 34.90 |
| STEP 2 | 54.90 | 3.25$\times$ | 45.10 |
| STEP 3 | 58.04 | 3.65$\times$ | 49.80 |

As shown in Table 11, applying beam search yields steady and significant improvements in ACC, SP, and OPT across steps, since each iteration selects and refines the best-performing candidate. This confirms that ECO integrates cleanly with beam-search exploration and can benefit from multi-step optimization without any methodological conflict.

Beam search, however, incurs additional computational cost because each step requires generating multiple candidates and evaluating their runtime. Compared with the ROI-Retriever–only setting in Table 3, which already achieves a 3.10× speedup in a single step, the first beam-search step is less effective and only surpasses it at later iterations. This is largely due to the structural constraint that beam search must operate in a 1-shot setting: once a retrieved pair is used to produce the best candidate, it cannot be reused, reducing retrieval diversity early on.

Overall, ECO and beam search provide complementary benefits. ECO delivers strong one-step improvements, while beam search enables further iterative gains, showing that combining the two can be advantageous when additional computation is acceptable.

