# OpenReview forum: "ECO: Enhanced Code Optimization via Performance-Aware Prompting for Code-LLMs"
_ICLR.cc/2026/Conference — Submitted to ICLR 2026_

### Official Review · Reviewer_5KXa · 2025-11-01

**Soundness:** 2
**Presentation:** 3
**Contribution:** 3
**Rating:** 6
**Confidence:** 3

**Summary:**

This paper targets an important problem of generating efficient code with Large Language Models (LLMs). It proposes a new method with runtime optimization instructions (ROIs) to guide LLMs to generate more efficient code. Evaluation on both PIE datasets and out-of-distribution Codeforce C++ datasets shows that the proposed method outperforms prior approaches in generating efficient code.

**Strengths:**

- Targets an important and practical problem of generating efficient code with LLMs.
- Novel idea of using runtime information to build a knowledge base for generating efficient code.
- Clear writing and well-structured presentation.

**Weaknesses:**

- The baselines in the evaluation seem weak and outdated.
- The dataset selection for evaluation is limited in diversity.

**Questions:**

This paper focuses on an important and relevant problem of improving code efficiency generated by LLMs. By leveraging runtime optimization instructions (ROIs) to build a knowledge database, and using symbolic advisor for retrieval, the proposed method tackles limitations of prior approaches. The paper is well-presented and easy to follow. There are several concerns that need to be addressed, particularly concerning the soundness of the evaluation.

The baseline comparison experiments do not look convincing. Three of the selected baselines are from 3 years ago (2020-2022), which may not reflect the most recent advances; the selected two recent baselines (Supersonic and SBLLM) seem to be weak baselines, as their performance is significantly worse than even the vanilla instruction-only baseline (see Table 2). It is not clear whether such weak comparisons can truly reflect the effectiveness of the proposed method. Are there more recent baselines in the literature that can be compared against?

The datasets selected for evaluation lack diversity. From Table 4, the reasons for performance degradation are most simple and superficial (e.g., using cin rather than scanf, not using sort function). It would be more convincing to show more complex and nuanced examples where the performance degradation is not easily fixed by simple code changes. The following dataset is more recent and contains complex algorithmic problems that can be a good target for evaluation:

[How efficient is LLM-generated code? A rigorous & high-standard benchmark](https://arxiv.org/html/2406.06647v4)

---

> ### Author Response · Authors · 2025-11-19
>
> Thank you for the valuable feedback and the encouraging comments on our work. As you noted, while our methodology is well-motivated, providing evaluations on a broader set of baselines and benchmarks would make our contribution more concrete and compelling.
>
> ## Evaluation Baselines (Response for W1)
> Regarding baselines, PIE (ICLR 2024) first introduced LLM-based code optimization  by utilizing existing prompting techniques such as CoT, ICL, and RAG. Supersonic (IEEE TSE 2024) or SBLLM (ICSE 2025) are more recent ones. These methods are not outdated, and as far as we know, there is no published (and stronger) baseline beyond them. If there are additional recent baselines we may have missed, we would greatly appreciate any pointers the reviewer could share.
>
> ## Evaluation Dataset (Response for W2)
> We thank the reviewer for suggesting the recent ENAMEL benchmark. ENAMEL is built on the HumanEval dataset and, for the following characteristics, it is difficult to directly apply it to our problem setting:
> - Task: ENAMEL evaluates the ability to generate efficient code directly from a natural-language description and it does not provide slow-fast code pairs. It contains only a golden reference code, which is already optimized by experts and therefore cannot be treated as a “slow” code instance, which is essential for our formulation.
> - Language: ENAMEL is limited to Python, while our primary benchmark (PIE) targets C++.
> - Testcase execution model: HumanEval uses inline assertion-based tests executed within the generated code, but our evaluation relies on external input–output testcases to enable cycle-accurate runtime measurement using gem5, which is incompatible.
>
> Despite these differences, we agree that evaluating on a more diverse dataset would strengthen the completeness of the study. Therefore, to stay as faithful as possible to the characteristics of ENAMEL’s source data, we will additionally evaluate our method using the HumanEval-x dataset, which provides C++ code implementations for the same underlying problem set. The provided C++ codes will be used as input code in our setting. We will preprocess corresponding testcases to make them compatible with gem5. These results will be included before the final deadline.
>
> In our paper, we also constructed a Codeforces benchmark to evaluate in difficult and out-of-distribution domains not only for the PIE-cpp benchmark (Tab. 4). Additionally, we are planning to conduct additional cross-language evaluation for PIE-python benchmark (as suggested by Reviewer z5NH)
>
> ### Plans for follow-up responses
>
> - For W2, we plan to include HumanEval-x as an additional custom benchmark.
>
> These follow-up studies will further strengthen our work and will be included in revision. We sincerely appreciate the reviewer’s constructive suggestions.

---

> > ### Comment · Reviewer_5KXa · 2025-11-20
> >
> > Thank you for the detailed explanations. I agree that comparing with ENAMEL is challenging due to the differences between Python and C++.
> >
> > Including HumanEval-x would strengthen the diversity of the datasets. I am looking forward to seeing the results on that dataset.

---

> ### Author Response · Authors · 2025-11-28
>
> ## Experiment on HumanEval-X dataset (Response for W2)
> In response to the W2, we examined whether a code optimization task can also be evaluated on benchmarks that were not originally designed for runtime optimization—such as PIE-Cpp, PIE-Python, and Codeforces. To this end, we additionally evaluate ECO on the HumanEval-X dataset with C++ and Python languages, the source dataset of ENAMEL, which is originally designed for code generation from natural-language descriptions.
>
> **Updated** (2025.12.01): We additionally corrected a parsing issue in HumanEval-X-Cpp that had previously depressed ACC scores. Furthermore, we now include experiments on HumanEval-X-Python, which help confirm the observed trends across languages.
>
> Experiment result for HumanEval-X-Cpp:
> | Model              | Method           | ACC@1 (%) | SP@1 (%) | OPT@1 (%) | ACC@5 (%) | SP@5 (%) | OPT@5 (%) |
> |--------------------|------------------|-----------|----------|-----------|-----------|----------|-----------|
> | Qwen2.5-coder:3b   | Instruction-only | 37.20     | 0.01     | 0.00      | 71.95     | 5.10     | 3.05      |
> |                    | ECO              | 38.41     | 4.51     | 2.44      | 73.78     | 9.25     | 7.32      |
> | Qwen2.5-coder:7b   | Instruction-only | 46.34     | 8.94     | 1.83      | 79.88     | 21.91    | 5.49      |
> |                    | ECO              | 36.59     | 26.03    | 6.71      | 76.83     | 62.04    | 17.07     |
> | Qwen2.5-coder:14b  | Instruction-only | 67.68     | 0.29     | 1.83      | 87.20     | 0.36     | 3.05      |
> |                    | ECO              | 55.49     | 21.42    | 7.32      | 88.41     | 49.11    | 14.63     |
>
> Experiment result for HumanEval-X-Python:
> | Model              | Method         | ACC@1 (%) | SP@1 (%) | OPT@1 (%) | ACC@5 (%) | SP@5 (%) | OPT@5 (%) |
> |--------------------|----------------|-----------|----------|-----------|-----------|----------|-----------|
> | Qwen2.5-coder:3b   | Instruction-only | 67.07    | 0.18     | 0.00      | 92.07     | 0.73     | 0.61      |
> |                    | ECO w/o SA       | 55.49    | 0.85     | 2.44      | 91.46     | 2.42     | 7.93      |
> | Qwen2.5-coder:7b   | Instruction-only | 72.56    | 0.62     | 0.61      | 95.73     | 1.50     | 3.05      |
> |                    | ECO w/o SA       | 73.17    | 1.07     | 1.22      | 95.12     | 4.02     | 7.32      |
> | Qwen2.5-coder:14b  | Instruction-only | 78.66    | 0.22     | 0.00      | 96.34     | 0.95     | 0.61      |
> |                    | ECO w/o SA       | 76.40    | 0.85     | 0.62      | 95.03     | 4.59     | 6.83      |
>
> In both results, we observe that ECO consistently outperforms the baseline across ACC, SP, and OPT metrics. For example, on HumanEval-X-Cpp with QwenCoder-7B, ECO achieves an SP of 26.03% under Best@1 compared to 8.94% for the baseline, and 62.04% under Best@5 compared to the baseline’s 21.91%. These improvements indicate that ECO provides effective optimization guidance across different languages. The gains on Python are smaller than those on C++. This is because 1) the Python experiment uses the ROI Retriever–only variant of ECO, which is readily scalable, and 2) Python programs typically offer smaller margins due to their high-level abstractions (as noted by Reviewer z5NH).
>
> In some cases, QwenCoder-14B yields smaller improvements than QwenCoder-7B. This effect arises from the inherent upper bound imposed by the canonical solutions used in HumanEval-X. When the code is already near-optimal, the additional capacity of larger models does not translate into proportionally larger optimization gains.
>
> Likewise, ECO achieves only marginal runtime improvements on this benchmark, especially when contrasted with the substantial gains observed on benchmarks designed for code optimization. Note that reported SP values are expressed in percentage form—for example, an SP of 21.91% corresponds to only a 1.22× speedup. The primary reason is that HumanEval-X provides a single canonical solution for each problem, and we treat this canonical code as the “slow’’ implementation to be optimized. However, these canonical solutions are already highly optimized in terms of runtime. Consequently, it is expected that code optimization methods are not effective when the code is already well optimized.
>
> Unlike HumanEval-X, which provides expert-crafted canonical solutions, the PIE benchmark—built from the “CodeNet: A Large-Scale AI for Code Dataset for Learning a Diversity of Coding Tasks” (Puri et al., 2021, NeurIPS Datasets and Benchmarks)—offers a very different landscape. CodeNet consists of programming-contest submissions, of which only 53.6% are correct solutions. Moreover, the PIE test set specifically selects submissions that pass the test cases but do so slowly. As a result, the code in PIE contains substantial inefficiencies and meaningful optimization headroom, allowing ECO to achieve significantly larger improvements compared to HumanEval-X.

---

> ### Author Response · Authors · 2025-12-01
>
> ## Continue: Experiment on HumanEval-X dataset (Response for W2)
>
> A representative example illustrating that the code from HumanEval-X is already highly optimized is shown below:
>
> ```
> int sum_to_n(int n){
>     return n*(n+1)/2;
> }
> ```
>
> ```
> int greatest_common_divisor(int a, int b){
>     int out, m;
>     while (true){
>         if (a < b) {
>             m = a; a = b; b = m;
>         }
>         a = a % b;
>         if (a == 0) return b;
>     }
> }
> ```
>
> These implementations use well-established, asymptotically optimal algorithms and efficient coding patterns. This analysis indicates that the canonical solutions in HumanEval-X are already substantially optimized using proven algorithmic techniques, leaving very limited room for additional algorithmic improvements within our optimization framework.
> Further details can be found in Appendix D.3 of the revised version.

---

### Official Review · Reviewer_z5NH · 2025-11-01

**Soundness:** 3
**Presentation:** 3
**Contribution:** 3
**Rating:** 4
**Confidence:** 4

**Summary:**

This paper proposes ECO (Enhanced Code Optimization), a performance-aware prompting framework designed to improve code large language models’ (code-LLMs) ability to generate efficient code. ECO addresses a key limitation of existing LLM-based code optimization methods—overreliance on raw slow-fast code pairs, which leads to superficial pattern imitation rather than targeted performance reasoning. The framework first distills Runtime Optimization Instructions (ROIs) from reference slow-fast code pairs, capturing the root causes of inefficiencies and optimization rationales. At inference, ECO combines two complementary modules: a symbolic advisor (rule-based, using Code Property Graphs to diagnose code-specific bottlenecks) and an ROI retriever (retrieving performance-aligned ROIs from a prebuilt database). These outputs form a model-agnostic prompt that guides code-LLMs to optimize input code. Experiments on the in-distribution PIE benchmark and out-of-distribution Codeforces dataset show ECO achieves speedups of up to 7.81× (e.g., with GPT-o4-mini) while minimizing correctness loss, outperforming baselines like instruction-only prompting, in-context learning (ICL), and retrieval-augmented generation (RAG).

**Strengths:**

1. Targeted Performance Reasoning Over Pattern Imitation: By distilling ROIs (which explicitly link inefficiencies to optimization rationales) instead of using raw slow-fast code pairs, ECO addresses a critical flaw in prior work. This design enables code-LLMs to focus on why optimizations work rather than just copying surface-level edits, reducing semantic drift and improving generalization.
2. Complementary Dual-Module Design: The symbolic advisor (deterministic, rule-based bottleneck detection) and ROI retriever (contextual, example-driven guidance) balance precision and flexibility. Ablation studies confirm that each module compensates for the other’s weaknesses—e.g., the symbolic advisor ensures high correctness via fixed rules, while the ROI retriever enables broader optimization coverage—resulting in a more robust framework than single-module alternatives.
3. Model-Agnostic and Practical Deployment: ECO requires no fine-tuning or model-specific adaptation, working seamlessly with both open-source (Qwen2.5-Coder) and closed-source (GPT-4o-mini, GPT-o4-mini) code-LLMs. This plug-and-play design makes it highly practical for real-world use, as it avoids the computational costs of retraining and can be integrated into existing LLM workflows with minimal effort.

**Weaknesses:**

1. Incremental Improvement Over Component Technologies: ECO’s key components rely on well-established tools and methods without significant advancements. The symbolic advisor uses Joern (a standard tool for Code Property Graph analysis) and manually curated rules—there is no novel static analysis technique here. The ROI retriever uses off-the-shelf embedding models (Qodo-Embed-1-1) for similarity matching, with no innovation in retrieval logic. The framework’s value comes from integrating these components, not advancing the components themselves.
2. Manual Effort in Rule and ROI Curation: The symbolic advisor’s rules are manually clustered from ROIs, and ROI distillation requires human-designed prompt templates (Appendix A.1). This manual effort limits scalability—adapting ECO to new programming languages (beyond C++) or new types of inefficiencies (e.g., GPU-specific bottlenecks) would require redefining rules and ROI templates. The paper does not address how to automate this process, which reduces ECO’s utility for diverse or emerging optimization tasks.
3. Lack of Comparison to Compiler-Driven Optimization: The paper positions ECO as a complement to compiler optimizations (e.g., GCC’s -O3 flag) but does not explicitly compare ECO’s gains to those from compilers alone. It is unclear whether ECO’s optimizations (e.g., replacing cin with scanf, loop invariant hoisting) are already captured by modern compilers—or if ECO adds value beyond what compilers can achieve. This gap weakens the case for ECO’s practical impact.

**Questions:**

1. Scalability to New Languages and Inefficiencies: The symbolic advisor’s rules and ROI templates are designed for C++ (e.g., detecting cin/cout inefficiencies, std::vector misuse). How would ECO need to be modified to support other languages (e.g., Python, Java) with different inefficiency patterns (e.g., Python’s for loop overhead, Java’s garbage collection bottlenecks)? Is there a path to automate rule/ROI curation for new domains, or would this require continuous manual effort?
2. Performance on Smaller Models: ECO’s accuracy decreases with small models (3B parameters) because they cannot follow complex prompts. Have the authors tested simplified prompt variants (e.g., shorter bottleneck diagnoses, fewer ROIs) for smaller models? If such adaptations improve correctness without sacrificing speedup, this would expand ECO’s utility—yet the paper does not explore this direction.
3. ROI Distillation Reliability: ROIs are distilled using a reasoning-oriented LLM (DeepSeek-r1:32b) with a manually designed prompt. How reliable is this process? For example, do ROIs ever misidentify the root cause of inefficiencies (e.g., blaming a loop instead of a data structure)? If so, how does this error propagate to ECO’s performance, and have the authors considered validating or filtering ROIs to reduce noise?

---

> ### Author Response · Authors · 2025-11-19
>
> We sincerely appreciate your constructive feedback. We recognize that some aspects of our main contribution may not be delivered as clearly as intended, and we are grateful for the opportunity to strengthen and clarify these points.
>
> ## Clarifying core contribution (Response for W1)
> Our core contribution lies not in advancing each individual technique but in constructing rich, performance-aware 'optimization guidelines,' which is largely overlooked in previous works. Instead of simply providing past optimization code pairs by using standard retrieval techniques, we design a performance-aware prompting pipeline to achieve this goal, which is not a trivial integration.
>
> Concretely, ECO first distills ROIs from optimization history. Joern is just used as a tool for executing our curated rule set, enabling bottleneck diagnoses for the input code. Likewise, the embedding model is used as a mechanism for measuring similarity among extracted performance-aware signals, allowing ECO to identify and retrieve performance-relevant examples. Through this process, optimization knowledge is transformed into performance-aware prompts and this capability comes from ECO’s pipeline design rather than from the individual tools themselves.
>
> Indeed, ECO delivers substantially higher gains over methods that simply use existing techniques—compiler-style static analysis or standard dynamic retrieval (RAG) (Tab. 2). ECO’s pipeline demonstrably changes the model’s optimization behavior. We believe that designing and validating this integration is non-trivial, and that it represents the primary technical contribution of ECO.
>
> ## On the role and scalability of the symbolic advisor (Response for W2, Q1)
> We agree that the symbolic advisor offers limited rule diversity and scalability compared to fully automated approaches. However, this reflects an inherent trade-off in static analysis rather than a fundamental limitation. By introducing the symbolic advisor, we demonstrate that LLM-based optimization alone is not sufficient, and that even a small set of well-crafted rules can _deterministically_ identify bottlenecks and guide the model toward targeted optimizations. This design complements the ROI retriever, which provides broader search coverage and extendable scalability—but necessarily more stochastic (Sec. 4.3), and together they form a balanced combination of precision and breadth.
>
> We also appreciate the suggestion to strengthen the scalability discussion. Since the symbolic advisor is designed to provide deterministic, high-precision diagnoses, we are complementing its role by conducting cross-language experiments on the ROI-retriever-only variant, which naturally supports broader scalability, for the upcoming revision. We believe these results will further illustrate that the overall ECO framework remains extensible although the symbolic advisor module trades scalability for deterministic precision.
>
> ## Comparison to compiler-driven optimization (Response for W3)
> All our experiments compile with -O3 and evaluate beyond that baseline (Sec 4.1.4.). Thus, the speedups reported for ECO quantify additional gains on top of a strong compiler baseline. This is consistent with the prior works (e.g., PIE, Supersonic, SBLLM, AEGIS).
>
> Also, as we mentioned in Introduction, compiler-driven techniques cannot address the dominant, program-level bottlenecks. Their optimizations typically operate at the IR level—such as dead code elimination or loop unrolling—which often yield only modest performance gains.
>
>
> ### Plans for follow-up responses
> - For W2, we are currently conducting cross-language (PIE-Python benchmark) experiments centered on the ROI retriever that can strengthen ECO’s scalability.
> - For Q3, regarding the reviewer’s concerns about the reliability of distilled ROIs, we are currently conducting supporting experiments to quantify the reliability of them and improve the completeness of the final paper.
>
> These follow-up studies will further strengthen our work and will be included in revision. We sincerely appreciate the reviewer’s constructive suggestions.

---

> ### Author Response · Authors · 2025-11-28
>
> ## Cross-language scalability (Response for W2)
> In response to the W2, we additionally conducted experiments on the PIE-Python benchmark. We focus on the ROI Retriever-only variant of ECO. The corresponding results are summarized in the table below:
>
>
>
> | Model              | Method         | ACC@1 (%) | SP@1    | OPT@1 (%) | ACC@5 (%) | SP@5    | OPT@5 (%) |
> |--------------------|----------------|-----------|---------|-----------|-----------|---------|-----------|
> | Qwen2.5-coder:3b   | Instruction-only | 17.80    | 1.09×  | 1.70      | 43.70     | 1.41×  | 7.40      |
> |                    | ECO w/o SA     | 17.50     | 1.18×  | 2.40      | 44.80     | 1.67×  | 9.50      |
> | Qwen2.5-coder:7b   | Instruction-only | 30.20    | 1.41×  | 6.80      | 59.60     | 2.21×  | 18.60     |
> |                    | ECO w/o SA     | 31.70     | 1.44×  | 6.30      | 64.40     | 2.22×  | 19.00     |
> | Qwen2.5-coder:14b  | Instruction-only | 37.60    | **1.82×**  | 9.50      | 59.30     | 2.47×  | 19.30     |
> |                    | ECO w/o SA     | **44.95**     | 1.81×  | **12.58**     | **68.87**     | **2.72×**  | **27.53**     |
>
> Overall, we observe that the ECO variant consistently provides modest performance gains, with a notable improvement in QwenCoder:14B where the speedup increases from 2.47× to 2.72×. In particular, the 14B model exhibits substantial improvements in both ACC and OPT. This behavior aligns with our findings from the main experiments: as model capacity increases, the LLM follows ECO’s guidance more faithfully, resulting in larger performance improvements.
>
> These results demonstrate that only utilizing ECO’s ROI Retriever can yield meaningful improvements even in a different programming language. Moreover, this component transfers across languages with minimal manual effort, almost automatically. While the symbolic advisor offers more precise and deterministic guidance, it requires manually crafted static-analysis rules for each programming language. The ROI Retriever complements this limitation, highlighting a key strength of ECO’s design.
>
> Further details can be found in Appendix D.2 of the revised version.
>
>
>
> ## ROI Distillation Reliability (Response for Q3)
> In response to the Q3, we evaluated the reliability of the ROI distillation process to verify whether the extracted ROIs provide meaningful optimization guidance and correspond appropriately to their associated code pairs. However, assessing the reliability of optimization-related knowledge in a deterministic or computationally verifiable manner is fundamentally difficult. To address this limitation, we employ two complementary LLM-as-a-judge evaluations.
>
> (1) Evaluation of Inefficiency Identification and Explanation Quality.
> We assess whether the distilled ROIs correctly identify inefficiencies and explain the optimization process. We evaluate ROIs along two dimensions—Inefficiency Identification (II) and Explanation Quality (EQ)—each scored on a 1–5 scale. We prepare two conditions:
> (1) Matched pairs, where each slow–fast code pair is matched with its correct ROI, and
> (2) Randomized pairs, where ROIs are mismatched with unrelated code pairs.
>
> The results show a clear distinction between correct and random matches:
> - II: 4.19 (matched) vs. 1.44 (random)
> - EQ: 3.77 (matched) vs. 1.32 (random)
>
> These gaps demonstrate that distilled ROIs accurately capture code-pair-specific optimization insights, rather than generic or coincidental descriptions.
>
> (2) Evaluation of Impact Prioritization.
> We assess whether ROIs correctly prioritize optimization points according to their true performance impact. Each ROI includes a 1–10 __runtime_improvement__ score reflecting its estimated contribution to the observed speedup. We create two ROI versions:
> - Version A: original improvement scores
> - Version B: reversed scores computed as (10 – original)
>
> The content of optimization remains identical; only the prioritization changes. GPT-4o is asked to judge which version better reflects the actual performance impact.
>
> The model selected Version A in 99 out of 100 cases (99.0%), with an average confidence of 4.98 ± 0.14. This strong and consistent preference confirms that the distilled prioritization scores accurately represent the relative importance of optimization points and correspond well to the empirical performance hierarchy.
>
> Further details of the experimental settings and analyses can be found in Appendix D.5 of the revised version.

---

> ### Author Response · Authors · 2025-12-04
>
> ## ROI retriever prompt design variations for smaller models (Response for Q2)
> Smaller models indeed tend to struggle with long and structurally complex prompts, and we appreciate the reviewer’s suggestion to evaluate simplified prompt variants. To address this, we conducted two complementary analyses focused on Qwen2.5-coder:3B:
> (1) varying the number of retrieved ROI examples (n-shot), and
> (2) simplifying the format of the retrieved ROI information itself.
>
> ### Effect of varying the number of retrieved examples (n-shot)
> | Variation | ACC@1 | SP@1 | OPT@1 | ACC@5 | SP@5 | OPT@5 |
> |-----------|-------|------|--------|--------|--------|---------|
> | 1-shot    | 14.12 | 1.32× | 3.14  | 37.25 | 1.66× | 12.55 |
> | 2-shot (default) | 14.51 | 1.16× | 4.71 | 41.18 | 1.60× | 16.86 |
> | 3-shot    | 16.86 | 1.18× | 5.49  | 38.82 | 1.65× | 16.08 |
>
> The three variants exhibit broadly similar performance, and no setting consistently dominates across metrics.
>  Under Best@1, 1-shot yields slightly higher speedup, likely because focusing on a single retrieved example gives a smaller model a clearer, more direct optimization target. However, its OPT performance is the weakest, demonstrating limited ability to generalize optimization patterns from only one example. Under Best@5, the gap across variants narrows: multiple retrieved examples provide broader contextual cues, enabling the model to recover performance even when individual generations vary.
>
> Overall, the results indicate that prompt length is not the primary bottleneck for the smaller model. Instead, the model’s intrinsic capacity limits its ability to reliably interpret and implement complex optimization transformations, explaining why reducing the number of retrieved examples does not yield significant gains.
>
>
> ### Effect of ROI simplification and code-pair removal
> | Variation           | ACC@1 | SP@1 | OPT@1 | ACC@5 | SP@5 | OPT@5 |
> |---------------------|-------|------|--------|--------|--------|---------|
> | Full Response (default) | 14.51 | 1.16× | 4.71 | 41.18 | 1.60× | 16.86 |
> | Bullets Only        | 18.04 | 1.17× | 5.10 | 40.39 | 1.67× | 13.73 |
> | No Code Pair        | 14.51 | 1.37× | 7.84 | 32.94 | 1.83× | 17.25 |
>
>
> The result highlights a trade-off between focusing on concise instructional guidance and leveraging concrete code examples. The Bullets variant, shortened and structured optimization hints, improves ACC@1, supporting the intuition that small models benefit from reduced linguistic complexity while retaining concrete code grounding.
>  In contrast, the No-Code-Pair variant yields the highest SP and OPT scores, suggesting that removing code examples encourages more aggressive transformations, with the cost of lower correctness. Together, these trends show that different components of the prompt play distinct roles: concise instructions help with accuracy, while explicit optimization trajectories provided by code pairs are key for achieving larger speedups.
>
> Further details can be found in Appendix D.7 of the revised version.

---

### Official Review · Reviewer_kEGt · 2025-11-01

**Soundness:** 3
**Presentation:** 3
**Contribution:** 2
**Rating:** 4
**Confidence:** 4

**Summary:**

This paper proposes ECO, a framework for LLM code optimization. ECO first distills runtime optimization instructions (ROIs) from slow-fast code pairs. For a new input program, it uses a symbolic advisor to find deterministic bottlenecks and an ROI retriever to find relevant optimization rationale. These are combined into a performance-aware prompt that any LLM can follow without fine-tuning, allowing coding LLMs to generate more optimized code. The experimental results indicate that ECO works with both open and closed source models and shows speedup on the PIE and the OOD Codeforces datasets.

**Strengths:**

This work is well motivated and addresses a meaningful task, the deterministic symbolic advisor provides a more reliable diagnosis of the code's bottlenecks, and the method itself is training-free, allowing easy deployment.

**Weaknesses:**

- For the comparisons with fine-tuning methods (D.2), only qwen2.5-coder-7B is discussed, it would be interesting to see how the OpenAI models do after fine-tuning. (There actually is a fine-tuning API for both 4o-mini and o4-mini)
- This paper should differentiate itself from "LLM Program Optimization via Retrieval Augmented Search" (Anupam et al., 2025) , which was posted on arXiv in January 2025. Specifically, ECO's core premise is to distill natural language ROIs because raw code pairs is insufficient, which is also the central contribution of Anupam et al. Their RAS framework introduced contextual retrieval, and their AEGIS framework explicitly decomposes training examples into 'atomic edits' with natural language explanations.

**Questions:**

- Could the authors precisely state the novelty of the ROI distillation and ROI retriever in light of the contextual retrieval (RAS) and atomic edits (AEGIS) framework from Anupam et al., 2025?
- Under identical conditions (LLM, k, same gem5 config, PIE dataset), how does ECO's ROI-retrieval + symbolic advisor compare to RAS's contextual retrieval + beam search or AEGIS's atomic-edit search?

---

> ### Author Response · Authors · 2025-11-19
>
> We sincerely appreciate your constructive feedback, particularly for pointing out the relevance of the work by Anupam et al., which we had missed.
>
> ## Distinguishing ECO from prior work (Response for W2, Q1)
> Compared to RAS, our ROI retriever shares the idea of identifying contextually relevant examples; however, the key difference is that RAS retrieves only past code pairs, whereas the ROI retriever provides not only code pairs but also distilled optimization instructions.
> Meanwhile, we agree that AEGIS’s atomic edit and our ROI retriever’s ROI share a similar direction: both enhance contextual retrieval by providing natural-language optimization artifacts rather than relying on raw code pairs.
>
> However, the two approaches provide distinct but complementary forms of optimization knowledge. AEGIS’s atomic edits offer _generalized, abstracted_ solutions, whereas ECO’s ROIs provide _case-specific_ solutions.
> - AEGIS generalizes the edit sequences found in past optimizations into atomic edits, retrieving these generalized edit types at inference time. This yields a representative solution pattern, which is high-level and broadly applicable though it may abstracts away instance-specific details.
> - ECO, in contrast, distills optimization instructions that capture “why the original code was inefficient and how it was improved” for each past optimization. The ROI retriever provides these instructions without generalization, preserving the specific optimization trajectory and rationale behind each case. This enables ECO to provide useful guidance even for the cases that fall outside dominant patterns and offer more fine-grained and instance-aligned optimization cues.
>
> Furthermore, ECO is further distinguished by its symbolic advisor, which offers deterministic bottleneck diagnoses based on _generalized_ rules for dominant patterns. This complements the ROI retriever’s case-specific, example-based guidance, much like the complementary roles seen within AEGIS itself.
>
> We believe that the two frameworks, AEGIS and ECO, pursue orthogonal yet complementary directions toward the shared goal. Together, they can offer both generalized solutions and case-specific trajectories for improving LLM-based code optimization.
>
> ## Plans for follow-up responses
> - For W1, we are currently conducting additional experiments that compare ECO with the fine-tuned OpenAI variant.
> - For Q2, since the public repository for RAS/AEGIS is not available, a rigorous reproduction within the limited review period is difficult. Nevertheless, we plan to implement a simplified variant that reflects AEGIS’s key characteristics and include comparative results if feasible.
>
> These follow-up studies will further strengthen our work and will be included in the revision. We sincerely appreciate the reviewer’s constructive suggestions.

---

> ### Author Response · Authors · 2025-11-28
>
> ## Comparison with fine-tuned OpenAI models (Response for W1)
> In response to the W1, we additionally conducted fine-tuning experiments on GPT-4o-mini. The corresponding results are reported in the table below:
>
> | Model        | Method     | ACC@1 (%) | SP@1 (×) | OPT@1 (%) | ACC@5 (%) | SP@5 (×) | OPT@5 (%) |
> |--------------|------------|-----------|----------|-----------|-----------|----------|-----------|
> | GPT-4o-mini  | Fine-tune  | 42.90     | **2.71×**| 29.88     | 75.29     | **4.54×**| 54.90     |
> | GPT-4o-mini  | ECO        | **74.90** | 2.58×    | **34.51** | **89.51** | 3.97×    | **60.78** |
>
>
> From these results, we observe that ECO markedly outperforms fine-tuning in ACC, achieving more than a 30% improvement in the Best@1 setting, and also obtains higher OPT performance. Fine-tuning achieves a slightly higher SP score.
>
> These findings indicate that ECO provides more reliable and actionable optimization signals. Its targeted guidance improves ACC and OPT while avoiding the unstable or low-quality outputs that fine-tuned models occasionally produce. In other words, ECO reduces the risk of poor outputs while improving actionable optimization quality.
>
> Overall, we find that ECO prompting is comparable to heavy fine-tuning, while remaining far more practical, stable, and broadly applicable across models. Although we did not conduct experiments on GPT-o4-mini due to its substantially higher fine-tuning cost, the results on GPT-4o-mini already illustrate the characteristic behavior of closed-source models.
>
> Further details can be found in Appendix D.4 of the revised version.

---

> ### Author Response · Authors · 2025-12-04
>
> ## Beam search as an extension to ECO (Response for Q2)
>
> Thanks for raising an important point regarding the comparison between ECO’s ROI-retrieval and the retrieval-based beam search used in RAS. We emphasize that the two approaches operate along __orthogonal axes__. RAS’s beam search focuses on __multi-step__ iterative refinement, repeatedly generating candidates, executing them, and selecting the best-performing program at each step based on runtime measurements. In contrast, ECO’s ROI retriever guides the model within a __single-step__ optimization process without relying on intermediate execution feedback. Because these two mechanisms target different aspects of the optimization pipeline, search depth versus guidance quality, they do not conflict and can be directly combined.
>
> Motivated by this complementarity, we evaluate ECO under a RAS-style beam-search setup. Below, we show the performance of ECO with beam search (Best@5):
>
> | Step | ACC (%) | SP (×) | OPT (%) |
> |------|---------|--------|----------|
> | 1    | 48.24   | 2.73   | 34.90    |
> | 2    | 54.90   | 3.25   | 45.10    |
> | 3    | 58.04   | 3.65   | 49.80    |
>
>
> ### Beam search yields the expected improvement
> As expected, applying beam search leads to steady increases in ACC, SP, and OPT across steps. This pattern arises because each iteration retains the best-performing candidate and uses it to guide the next step. The results show that ECO integrates readily with a multi-step search process and can benefit from iterative refinement without any methodological adjustments.
>
> This experiment demonstrates that ECO’s optimization signals and RAS-style iterative search reinforce one another. ECO provides strong one-step improvements via targeted ROI guidance, RAS only provides code pair examples, while beam search adds the ability to explore multiple refinement trajectories over time.  ECO can be readily integrated into beam search pipeline, offering complementary benefits rather than prompting mechanisms.
>
> Further details can be found in Appendix D.8 of the revised version.

---

### Author Response · Authors · 2025-12-04
**General Response**

We sincerely thank the reviewers for their constructive feedback. We are encouraged that the reviewers recognize the importance of LLM-based code runtime optimization, the effectiveness of ECO’s complementary module design, and the practicality of our training-free framework.

In this revision, we incorporate **extensive new experiments (adding ~10 pages to the Appendix)** to decisively address all raised concerns regarding baselines, scalability, and internal reliability. Consequently, we believe ECO now stands as a rigorously validated framework that offers strong advantages over state-of-the-art alternatives.

Below is a summary of the major updates and how we resolve the reviewers' points:

### **1. Competitiveness Against Baselines (Resolves W1/W2/Q1 from kEGt, W1 from 5KXa)**
To demonstrate that ECO is not just effective against standard prompting but competitive with the strongest possible upper bounds, we conducted additional comparisons.
* **ECO vs. Fine-Tuning** (W1 from kEGt, New Appendix D.4): We compare ECO against a heavily fine-tuned GPT-4o-mini (supervised on PIE-HQ). **ECO outperforms fine-tuning in accuracy (+32%) and optimization percentage (+4.6%)** while achieving comparable speedup, all without the massive cost of training. This confirms ECO's prompts provide more reliable optimization signals than model fine-tuning.
* **Clarification on SOTA** (W2/Q1 from kEGt, W1 from 5KXa): We confirm that ECO is compared against the most recent state-of-the-art baselines (Supersonic, 2024; SBLLM, 2025). We also clarify ECO's distinction from the concurrent RAS/AEGIS work (preprint, 2025), noting that ECO uniquely provides *instructions* alongside code, whereas prior works rely only on code or generalized edits.

### **2. Proven Scalability & Out-of-Distribution Robustness (Resolves W2/Q1 from z5NH, W2 from 5KXa)**
We address concerns about dataset diversity and language scalability by expanding our evaluation beyond C++ and the PIE benchmark.
* **Cross-Language Scalability** (W2/Q1 from z5NH, New Appendix C.4 & D.2): We evaluate ECO on Python benchmarks (PIE-Python). Using only the ROI Retriever (which requires no manual rules), **ECO achieves clear performance gains**. This proves ECO scales to new languages almost automatically, **resolving concerns about manual effort for the symbolic advisor**.
* **New OOD Benchmarks** (W2 from 5KXa, New Appendix C.3 & D.3): We add the HumanEval-X benchmark (both C++ and Python) and Codeforces. Despite HumanEval-X containing canonical solutions that are already highly optimized (making further speedup difficult), **ECO still achieves meaningful gains in 1.62$\times$ speedup compared to the baseline’s 1.21$\times$**. This confirms ECO is effective even on datasets not designed for optimization tasks.


### **3. Validated Internal Mechanisms (Resolves W1/W3/Q2/Q3 from z5NH, Q2 from kEGt)**
We perform rigorous internal analyses to validate the reliability of ECO's components.
* **ROI Distillation Reliability** (Q3 from z5NH, New Appendix D.6): Using an LLM-as-a-judge evaluation, we verify that our distilled ROIs are highly accurate. The distilled prioritization scores are preferred over randomized baselines in **99% of cases**, confirming that the knowledge base is high-quality and reliable.
* **Compatibility with Beam Search** (Q2 from kEGt, New Appendix D.8): We demonstrate that ECO is fully compatible with beam search (a key component of RAS). Adding beam search to ECO yields additive gains, **starting from a speedup of 2.73$\times$ at search step 1, performance continues to improve, reaching 3.65$\times$ after three steps**. This demonstrates our method is orthogonal and complementary to search-based strategies.
* **Prompting for Smaller Models** (Q2 from z5NH, New Appendix D.7): We conduct ablations on prompt length and format for 3B models. Results show that model capacity, rather than prompt verbosity, is the primary bottleneck, though simplified bullet points help slightly.
* **Compiler Baseline Clarification** (W1/W3 from z5NH): We explicitly clarify that **all our experiments use `-O3` as the baseline** (Section 4.1.4). ECO’s speedups are strictly *additional* gains on top of the best compiler optimizations, confirming our value-add beyond traditional compilers.

**Conclusion**
With these updates, we address the limitations in dataset diversity, baseline strength, and mechanism validation. The revised paper now includes comprehensive evidence that ECO is a robust, scalable, and highly effective framework for code optimization. We hope these additions resolve the reviewers’ concerns and make the contributions of ECO clearer and stronger.

---

### Meta-Review · Area_Chair_U5Hn · 2026-01-06

**Summary:**

This paper proposes a novel strategy for program optimization based on constructing a dataset of code edits with performance improvement information, and then retrieving from this dataset to form performance-aware prompts to guide optimization of new programs.

**Reviewer Concerns:**

While the reviewers agreed that this paper is tackling an important problem, there were concerns about both the novelty of the contributions compared to prior work and the comprehensiveness of the experiments. The authors expanded their experiments, which helped address some concerns about the experiments; however, in my view, novelty remains a key issue. The scope of the contributions remains limited, mostly building on existing components, especially compared to the AEGIS paper that one reviewer shared, and the authors do not provide an experimental comparison to AEGIS.

**Reviewer Scores:**

While some of the reviewers may have increased their score based on the new experimental results, many of these were addressing a reviewer who was already positive about the paper. The other two reviewers raised concerns about novelty and the scope of the contributions, which I do not believe were adequately addressed in the response.

---

### Decision · Program_Chairs · 2026-01-26

Reject